# State space models can express $n$-gram languages

**Vinoth Nandakumar**                                    *vinoth.90@gmail.com*
*Sydney AI Centre, University of Sydney*

**Qiang Qu**                                             *vincent.qu@sydney.edu.au*
*Sydney AI Centre, University of Sydney*

**Peng Mi**                                              *mipeng01@gmail.com*
*Sydney AI Centre, University of Sydney*

**Tongliang Liu**                                        *tongliang.liu@sydney.edu.au*
*Sydney AI Centre, University of Sydney*

**Reviewed on OpenReview:** *https://openreview.net/forum?id=QlBaDKb37O*

## Abstract

Recent advancements in recurrent neural networks (RNNs) have reinvigorated interest in their application to natural language processing tasks, particularly with the development of more efficient and parallelizable variants known as state space models (SSMs), which have shown competitive performance against transformer models while maintaining a lower memory footprint. While RNNs and SSMs (e.g., Mamba) have been empirically more successful than rule-based systems based on $n$-gram models, a rigorous theoretical explanation for this success has not yet been developed, as it is unclear how these models encode the combinatorial rules that govern the next-word prediction task. In this paper, we construct state space language models that can solve the next-word prediction task for languages generated from $n$-gram rules, thereby showing that the former are more expressive. Our proof shows how SSMs can encode $n$-gram rules using new theoretical results on their memorization capacity, and demonstrates how their context window can be controlled by restricting the spectrum of the state transition matrix. We conduct experiments with a small dataset generated from $n$-gram rules to show how our framework can be applied to SSMs and RNNs obtained through gradient-based optimization.

## 1 Introduction

Over the past two decades, breakthroughs in deep learning have revolutionized natural language processing tasks, including language modeling (Radford et al. (2018), Radford et al. (2019)), question answering (Devlin et al. (2018), Yang et al. (2019)), and machine translation **?**. While $n$-gram models were used widely for next-word prediction before the advent of deep learning, it was empirically observed that language models using recurrent neural networks (RNNs) were more effective for complex tasks. Although transformer models, with their attention-based layers, have since led to significant advancements in these tasks Vaswani et al. (2017), recent research has revisited RNN architectures Beck et al. (2024); Feng et al. (2024); Peng et al. (2023), and investigated variants known as state space models (SSMs). Notably, SSMs Gu et al. (2022; 2021); Wang & Xue (2024); Gu & Dao (2024) can be trained efficiently in parallel like transformers, while requiring less memory than transformers for inference.

Recent advances have largely been empirical, with new deep learning architectures developed through experimentation and validated on large datasets Liu et al. (2019); Radford et al. (2018; 2019). A key open problem in natural language processing is to provide a rigorous theoretical framework for the success of deep learning models on next-word prediction tasks. Such a framework could lead to the design of more efficient

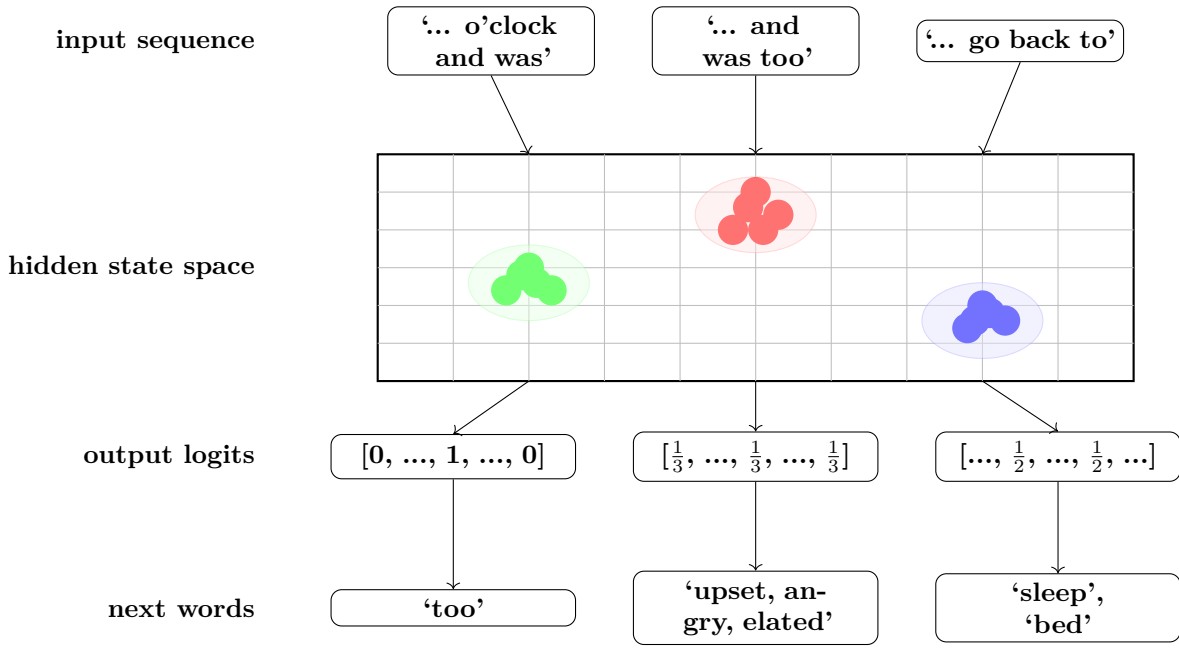

Figure 1: This diagram illustrates our framework for encoding $n$-gram rules with state space models. For example, the blue dots in the hidden state correspond to input sequences ending in "... go back to". The hidden state embedding vectors have been projected to two-dimensional space, and the vectors corresponding to the same $n$-gram form clusters. The output logits represent the next-word probabilities, and here the non-zero values correspond to the words that can occur next in the sequence (see Figure 2 for more details about this example of an $n$-gram model, and Figure 4 for a more detailed cluster plot).

architectures for SSMs and RNNs, and provide additional insights into the inner workings of these black-box models. A theoretical understanding of how SSMs and RNNs obtained through gradient-based optimization techniques encode the rules governing $n$-gram languages, and solve the next-word prediction task in this context, would be a first step towards a more rigorous mathematical framework explaining their success in natural language processing.

To theoretically explain the empirical success of these deep learning models, researchers have drawn upon mathematical models based on languages defined by formal grammars Chomsky (1956; 1959); Chomsky & Schützenberger (1963) and universal approximation theory Wang & Xue (2024). While simple examples of formal languages like $a^n b^n$, parity languages, and boolean expressions have been studied with neural networks in Gers & Schmidhuber (2001); Sennhauser & Berwick (2018); Hahn (2020), these are often too restrictive to capture the nuances of natural language. Another key result is the Turing completeness of RNNs Chung & Siegelmann (2021), which implies that they can recognize recursively enumerable languages. However, the proof uses weights that have unbounded precision (or unbounded memory), which does not explain their success in practical applications. Our work is closely related to Sarrof et al. (2024), which theoretically demonstrates that state space models can express regular grammars, but does not examine the depth and size required to model languages generated by $n$-gram rules.

In this paper we study the question of constructing state space models that express languages defined via $n$-gram rules from a theoretical viewpoint. Given a $n$-gram language model defined for a fixed language, we construct a state space model that represents it and bound its size (see Theorem 4.1 for a more precise statement). Our construction provides additional insight into the interpretability of SSMs and RNNs, and in particular the role of the state transition matrix and the neurons in the hidden layer. Our proof technique is illustrated in Figure 1 above. The contributions of this work can be summarized as follows.

- Our first key insight is that, in the context of $n$-gram models, next-word prediction is equivalent to memorizing the $n$-gram rules. Based on the observation that the embedding layer transforms the sequence of input words into a sequence of vectors, we reduce the problem to demonstrating that SSMs can memorize a set of input-output vectors under mild assumptions.

- Our key technical result provides a theoretical analysis of the memorization capacity of SSMs, extending similar findings in feedforward networks (see Yun et al. (2019)). This result is of independent interest and offers deeper insights into their behaviour.

- We analyze the context window of SSMs, particularly focusing on when the network's output depends solely on the most recent $n$ inputs. We demonstrate that this behaviour is closely related to the spectral decomposition of the state transition matrix.

- We conduct experiments with toy data consisting of English sentences generated from $n$-gram languages, and show how our theoretical insights can be applied to SSMs obtained via gradient-based optimization. We also discuss how our framework can be extended to RNNs, which incorporate a non-linearity into the hidden state update.

The paper is organized as follows. In Section 2, we describe related work on the theoretical underpinnings of language modelling with neural networks. In Section 3, we recall the definition of $n$-gram language models and state space models. In Section 4, we state the main theoretical results that show state space models are at least as expressive as $n$-gram language models, and outline the proof. In Section 5, we apply our theoretical framework to SSMs and RNNs that are trained with gradient-based optimization on a simple dataset from $n$-gram rules. In Sections 6, we discuss further directions based on our theoretical results. In the Appendix, we present detailed proofs of the main results.

## 2  Related work

**Comparison between neural networks and $n$-gram models.** The experiments in Bengio et al. (2003) using the Brown corpus and the Associated Press News corpus show that language models based on feedforward neural networks achieve lower perplexity scores compared to n-gram models (here the perplexity metric quantifies the model's ability to predict the next word in a sequence). In Chelba et al. (2017), empirical comparisons between RNNs and n-gram models have demonstrated the superior performance of RNNs for next-word prediction, based on experiments conducted using the UPenn TreeBank corpus and the One Billion Words benchmark. In Svete & Cotterell (2024), the authors theoretically analyze how transformer models can solve next-word prediction for datasets generated from $n$-gram rules, but their choice of query-key matrices only retain information about the positional encoding and not the word embedding, limiting their applicability to models trained using gradient-based optimization. Our work complements these empirical findings by providing a rigorous theoretical framework explaining how SSMs encode $n$-gram rules, which can be used to derive insights about the hidden layers and weights of SSMs and RNNs trained on simple datasets with stochastic gradient descent.

**Memorization and learning.** While memorization refers to the model's ability to fit pre-specified input-output pairs, learning involves capturing the underlying patterns or distributions present in the data, enabling the model to make accurate predictions on new, unseen inputs. The differences between memorization and learning have been analyzed through an empirical lens with gradient-based optimization in Arpit et al. (2017). The memorization capacity of feedforward neural networks has been studied theoretically in Yun et al. (2019); Zhang et al. (2017); Huang & Huang (1991), but their work does not extend to SSMs or RNNs. Our work builds on these theoretical results by analyzing the memorization capacity of SSMs, and illustrates how memorization enables deep learning models to encode $n$-gram rules for next-word prediction.

**On the expressiveness of RNNs and SSMs.** In Wang & Xue (2024), it is shown SSMs with layer-wise nonlinear activation can approximate any continuous sequence-to-sequence function with arbitrarily small error, building on analogous results for RNNs. In Jelassi et al. (2024), the authors theoretically analyze the effectiveness of transformers and SSMs on copying tasks. It has also been theoretically demonstrated that both RNNs and transformers are Turing complete, meaning they have the computational power to

simulate any Turing machine given sufficient time and resources Pérez et al. (2021); Chung & Siegelmann (2021). The connection between RNNs and formal languages can be viewed through the lens of the Chomsky hierarchy, Chomsky (1956), which classifies formal languages into four categories — regular, context-free, context-sensitive, and recursively enumerable — based on their generative power. RNNs, due to their Turing completeness, can theoretically recognize languages across this entire hierarchy Chung & Siegelmann (2021), but the construction there uses weights that have unbounded precision or unbounded memory, which are not realistic in practice. Our work complements these studies by presenting a theoretical framework showing how SSMs can model languages generated from $n$-gram rules without imposing any unrealistic assumptions, and we show how our theory can be applied to models trained via gradient-based optimization on simple datasets.

**Learning formal languages with stochastic gradient descent.** There has been a line of work using special cases of formal languages to investigate how they can be learnt with stochastic gradient descent using commonly used neural network architectures, such as LSTMs and transformers Ackerman & Cybenko (2020); Liu et al. (2023). The effectiveness of LSTM models on the languages $a^n b^n$ (and variants thereof) was established in Gers & Schmidhuber (2001), Bodén & Wiles (2000), while similar results for Dyck languages were obtained in Sennhauser & Berwick (2018). In the latter context, LSTM models require exponential memory in terms of the input length, and do not learn the underlying combinatorial rules perfectly. Analogous experiments with Dyck languages were carried out with transformer-based models in Ebrahimi et al. (2020), and the inner workings of these models were studied. However, the simple examples of formal languages studied in these papers do not model the English language effectively, and are less realistic than languages based on $n$-gram rules. While an extensive empirical evaluation was conducted in Delétang et al. (2023); Poel et al. (2024) by training a variety of neural network models on more complex formal languages, a theoretical understanding of these models is lacking. Our work complements these studies by presenting a theoretical framework showing how SSMs and RNNs can encode $n$-gram rules, as we demonstrate with experimental results using stochastic gradient descent.

## 3 Preliminaries on language modelling.

In this section we recall the definitions of $n$-gram languages and state space models, describing how they can be used for next-word prediction. We then give an overview of the main results in our work.

### 3.1 Language models

In this section, we formalize the language modelling task, which has been studied extensively in empirical work Radford et al. (2018), Radford et al. (2019) (it is often referred to as "text generation" or "text completion"). In empirical settings, the language $\mathcal{L}$ typically consists of sentences about a given set of topics that are semantically correct. We follow the conventions in Svete & Cotterell (2024).

**Definition 3.1.** The *vocabulary* $\mathcal{W}$ is a finite set consisting of all occurring words. Let $d = |\mathcal{W}|$, and $\mathcal{W}^* := \mathcal{W} \cup \{\emptyset\}$. For a given vocabulary $\mathcal{W}$, a *language* $\mathcal{L}$ is a subset of the set of sequences of words from $\mathcal{W}$, and represents the set of all possible sentences. Here the integer $N$ is the maximal length of a sentence.

$$\mathcal{L} \subset \bigsqcup_{0 \leq n \leq N} (\mathcal{W}^*)^n \qquad \blacksquare$$

**Definition 3.2.** Let $\underline{w} = (w_1, \cdots, w_k) \in \mathcal{W}^k$ be a sequence of words. Given $1 \leq i < j \leq k$, define the truncated sequence $\underline{w}_{i:j} = (w_i, \cdots, w_j)$. We say that $\underline{w} \in \mathcal{W}^k$ is a *valid starting sequence* if there exists another sequence $(w_{k+1}, \cdots, w_l)$ such that $(w_1, \cdots, w_k, w_{k+1}, \cdots, w_l) \in \mathcal{L}$. Let $\mathcal{L}_k \subset \mathcal{W}^k$ denote the set of all valid starting sequences with length $k$. $\blacksquare$

Given an input sequence $(w_1, \cdots, w_k)$, we can use a language model $f$ to generate the next word $w_{k+1}$; this process can be iterated to generate an output sentence $(w_{k+1}, \cdots, w_N)$ (see Goodman (2001), Katz (1987)). In the below definition, we assume a language model takes as an input a valid starting sequence, as defined above, and its output is a probability distribution that can be used to determine which words are likely to appear next in the sequence.

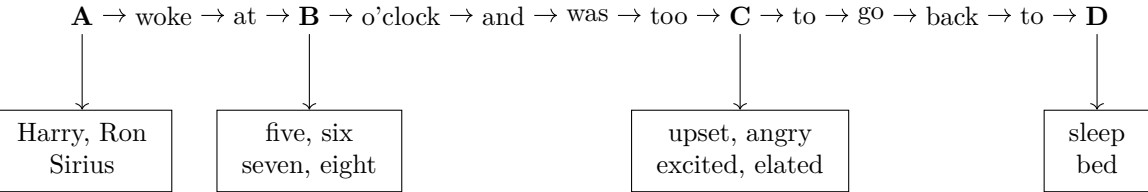

Figure 2: An example of an language $\mathcal{L}$ that can be modelled by $n$-grams, based on text from J.K. Rowling's "Harry Potter". The graph illustrates how sentences are formed, with each of the symbols A, B, C, D being replaced by one of the words in the corresponding box. Two examples of sentences from this language are "Ron woke at seven o'clock and was too upset to go back to bed." and "Sirius woke at five o'clock and was too elated to go back to sleep." See Appendix B for an expanded example, including a full list of $n$-gram rules.

**Definition 3.3.** Given a vocabulary $\mathcal{W}$, a *language model* is a function $f$ that takes in an arbitrary sequence $\underline{w} \in \mathcal{W}^k$, for some $k$, and outputs a vector of probabilities $f(\underline{w}) = (\hat{f}(w|\underline{w}))_{w \in \mathcal{W}^*}$. Here $\hat{f}(w|\underline{w})$ denotes the probability that the model $f$ will output the word $w$ given the input sequence $\underline{w}$.

### 3.2 $n$-gram language models

In this section, we rigorously define *n-gram language models* that we will use in subsequent sections. An $n$-gram language model generates sequences by estimating the probability of each word based on the $n-1$ preceding words, and capture local dependencies within a sequence Katz (1987); Goodman (2001). These models, widely used in natural language processing, can approximate real-world sentences but struggle with long-range dependencies Goldberg & Orwant (2013). They are closely related to the theory of formal grammars, which provide a more general mathematical framework for generating strings based on a set of combinatorial rules Chomsky (1956; 1959); Chomsky & Schützenberger (1963).

**Definition 3.4.** Given a language $\mathcal{L}$, an *n-gram language model* $f_{ng}^*$ consists of the set $(\mathcal{P}, f_{ng})$ (the integer $n$ is also fixed).

- Let $\mathcal{P} \subset (\mathcal{W}^*)^{n-1}$ be the set of all sequences of $n-1$ consecutive words appearing in $\mathcal{L}$.

- Let $f_{ng} : \mathcal{P} \to \Delta^d$ be a function, where $\Delta^d = \{x \in [0,1]^{d+1} | \sum_{i=1}^{d+1} x_i = 1\}$ is the $d+1$-dimensional probability simplex. The additional $d+1$-th entry corresponds to an unknown token $\emptyset$.

Given the above data, the $n$-gram language model $f_{ng}^*$ can be defined as follows (here $\underline{w} \in \mathcal{W}^k$ is any starting sequence). We use the $n$-gram assumption, which states that the probability of a given word $w$ occurring after a sequence $\underline{w}$ depends only on the last $n-1$ words in the sequence $\underline{w}$. If the last $n-1$ words $\underline{w}_{k-n+1:k-1}$ are not in $\mathcal{P}$, the output $v_\emptyset$ corresponds to the unknown token $\emptyset$.

$$f_{ng}^*(\underline{w}) = \begin{cases} f_{ng}(\underline{w}_{k-n+1:k-1}) & \text{if } \underline{w}_{k-n+1:k-1} \in \mathcal{P}, \\ v_\emptyset & \text{otherwise,} \end{cases} \qquad \blacksquare$$

**Example.** Here we give a simple example $n$-gram language model for the language $\mathcal{L}$ specified in the above Figure, with $n = 4$. For each phrase in $\mathcal{P}$, all valid choices for the next word are specified by the graph above, and we stipulate that they are all equally likely. The below formulas show how the function $f_{ng}$ is defined. The sequence ("and", "was", "too") is in $\mathcal{P}$, and the vector $v_{\text{"upset"}}$ denotes the one-hot vector in $\mathbb{R}^{d+1}$ corresponding to the word "upset".

$$f_{ng}(\text{"and", "was", "too"}) = \frac{1}{4}(v_{\text{"upset"}} + v_{\text{"angry"}} + v_{\text{"elated"}} + v_{\text{"excited"}})$$

$$f_{ng}(\text{"go", "back", "to"}) = \frac{1}{2}(v_{\text{"sleep"}} + v_{\text{"bed"}}) \qquad \blacksquare$$

We now introduce a notion of equivalence between language models below, using the total variation distance between two probability distributions. This extends the definition of "weak equivalence" from Section 2 of Svete & Cotterell (2024) (their definition corresponds to the $\epsilon = 0$ setting here). Below we fix a language $\mathcal{L}$ over a vocabulary $\mathcal{W}$.

**Definition 3.5.** Given a vocabulary $\mathcal{W}$ and a language $\mathcal{L}$, let $g$ be an $n$-gram language model, and let $f$ be a language model. For a fixed $\epsilon > 0$, we say that $f$ and $g$ are $\epsilon$-*equivalent when restricted to* $\mathcal{L}$ if the following condition holds for each valid sequence $\underline{w} \in \mathcal{L}_k$ with $k \geq n - 1$ (here note that $f(\underline{w}), g(\underline{w})$ are vectors in $\mathbb{R}^{d+1}$ and $|| \cdot ||_1$ denotes the $L_1$ norm of a vector).

$$||f(\underline{w}) - g(\underline{w})||_1 < \epsilon \qquad \blacksquare$$

### 3.3 State space language models

We start by recalling the definition of a state space model with a single hidden layer, following Section 4 of Wang & Xue (2024). Below $\sigma$ denotes the ReLU function applied to each coordinate of the corresponding vector. While it is similar to a recurrent neural network, in this setting the $\sigma$ non-linearity is applied while computing the output, and not during the hidden state update. We note that while the bias term $b_y$ is often not used in state space models, here we add it to the output to increase the expressiveness of the model.

**Definition 3.6.** A *state space model* $\mathcal{S}$ consists of an input layer, a hidden layer, and an output layer. The network is characterized by $n_0$ input neurons, $n_h$ hidden neurons, and $n_L$ output neurons. The weight matrices are denoted by $A$, $B$, and $C$, where $A \in \mathrm{Mat}_{n_h, n_h}(\mathbb{R})$ is the state transition matrix, $B \in \mathrm{Mat}_{n_0, n_h}(\mathbb{R})$ is the input-to-hidden matrix, and $C \in \mathrm{Mat}_{n_h, n_L}(\mathbb{R})$ is the hidden-to-output matrix. We omit the input-output matrix $D$, which is sometimes used to implement a skip connection.

It gives rise to a function $f_{\mathcal{S}} : (\mathbb{R}^{n_0})^T \to (\mathbb{R}^{n_L})^T$; the state space model processes a sequence of inputs $(x_1, x_2, \ldots, x_T)$, produces a sequence of hidden states $(h_1, h_2, \ldots, h_T)$ and yields outputs $(y_1, y_2, \ldots, y_T)$ via the following equations. Here $x_t \in \mathbb{R}^{n_0}$ is the input vector, $h_t \in \mathbb{R}^{n_h}$ is the hidden state vector, $y_t \in \mathbb{R}^{n_L}$ is the output vector, and $b_y$ is a bias term.

$$h_t = Ah_{t-1} + Bx_t \quad y_t = C\sigma(h_t + b_y) \quad \blacksquare$$

We recall how state space models can be used for next-word prediction in language modelling tasks (see also Section 4.3 of Gu et al. (2022)). The two key components that we need are a state space model and an embedding layer. The input to the state space language model is a sequence of words, so we start with a preprocessing step that converts a sequence of words to a sequence of one-hot vectors. The dimension of the vector $\theta^*(\underline{w})$ is proportional to the number of words in $\mathcal{W}$, which is usually quite large. In order to reduce the dimension of the vector before feeding it into the state space model, we define an embedding layer as follows.

**Definition 3.7.** Let $\bar{i} : \mathcal{W}^* \to \{0, 1, \cdots, d\}$ be a bijective map such that $i(\emptyset) = 0$, where $d = |\mathcal{W}|$. We define the word representation $\theta : \mathcal{W}^* \to \mathbb{R}^{d+1}$ as the map which sends $w \in \mathcal{W}^*$ to the corresponding one-hot vector in $\mathbb{R}^{d+1}$ (i.e. with a 1 in the $i(w)$-th position and 0-s elsewhere). Given an embedding matrix $W_0 \in \mathrm{Mat}_{e, d+1}(\mathbb{R})$ with distinct columns, denote by $\phi_{W_0} : \mathbb{R}^{(d+1)} \to \mathbb{R}^e$ the induced function. The embedding function $E$ is defined below; we say that the embedding layer has dimension $e$.

$$\underline{w} = (w_1, \cdots, w_N); \qquad E(\underline{w}) = [\phi_{W_0}\theta(w_1) \mid \cdots \mid \phi_{W_0}\theta(w_N)] \qquad \blacksquare$$

After the preprocessing step using the embedding layer, which sends $\underline{w}$ to $E(\underline{w})$, the resulting vector can in turn be fed into a state space model, which then outputs a vector of probabilities by applying a softmax layer at the end.

**Definition 3.8.** A state space language model $f_{\mathcal{S}}^*$ is a function obtained by combining an embedding layer $E$ as above, a state space model $\mathcal{S}$, and a softmax layer. Here $\mathcal{S}$ has $e$ input neurons, $d + 1$ output neurons and one hidden layer. It takes as input a sequence $\underline{w} = (w_1, \cdots, w_k)$ with variable length $k$, and outputs a probability vector in $\mathbb{R}^{d+1}$.

$$f_{\mathcal{S}}^*(\underline{w}) = \mathrm{softmax} \circ f_{\mathcal{S}} \circ E(\underline{w}) \qquad \blacksquare$$

# 4 Main results

In this section, we state the main results state space language models are more expressive than $n$-gram language models. In our proof, we construct a state space language model that encodes the $n$-gram rules which govern next-word prediction in this setting, through a rigorous theoretical analysis of their memorization capacity. We show how the context window of the state space model can be controlled by imposing restrictions on the spectrum of the state transition matrix. We also discuss how these results can be extended to recurrent neural networks, which incorporate a non-linearity into the hidden state update. We refer the reader to Appendix A for detailed proofs of all results.

## 4.1 Statement: overview

We present a framework for showing why state space models can represent any $n$-gram language model, with arbitrarily small error; our main result is as follows.

**Theorem 4.1.** *Let $\mathcal{L}$ be a language over a vocabulary $\mathcal{W}$. Let $f_{ng}^*$ be an $n$-gram language model, obtained from the datum $(\mathcal{P}, f_{ng})$. Given any $\epsilon > 0$, there exists a state space language model $f_{\mathcal{S}}^*$ with the following properties:*

- *The state space language model, $f_{\mathcal{S}}^*$ and $n$-gram language model $f_{ng}^*$ are $\epsilon$-equivalent when restricted to $\mathcal{L}$.*

- *The state space model component $\mathcal{S}$ can be chosen so that it has a single hidden layer with $|\mathcal{P}|$ neurons (here $|\mathcal{P}|$ is the number of $n$-gram rules in the language $\mathcal{L}$).*

- *The embedding layer has dimension $e$, where $e \geq 1$ is any fixed integer.* ∎

In the next section, we state a more precise version of the above Theorem which specifies the properties that the state space model must satisfy in order for the above to hold. We note that languages generated from $n$-gram rules are regular languages that can be represented by a finite state automata. In Theorem 4 of Sarrof et al. (2024), it is shown that state space models can represent any star-free regular language using a different approach based on Krohn-Rhodes theory, but without explicit bounds on the depth and size of the model required (see also Liu et al. (2023) for constructions of transformers that represent regular languages). Another line of work establishes the Turing completeness of RNNs Pérez et al. (2021); Chung & Siegelmann (2021), but their networks use either unbounded precision or unbounded memory, which limits their application to real-world settings.

Our approach does not have these limitations, and in the next section we explain how our insights can be translated to RNNs and state space models trained on toy English datasets. We note crucially that the number of neurons in the state space model does not depend on the number of sentences in the language $\mathcal{L}$, but only on the number of rules in the $n$-gram language $\mathcal{L}$. For any language $\mathcal{L}$, we can construct a model that solves next-word prediction by simply memorizing all of its sentences, but such a model would not provide much insight into learning, and the number of neurons would scale exponentially with the number of sentences in $\mathcal{L}$. The key feature of the above result is that the model does not simply memorize all of the input data, but rather learns the patterns in the data that are specified in the $n$-gram language.

We note that our statement above uses an $\epsilon > 0$ due to the presence of a softmax layer. The probability distribution obtained from a state space model with a softmax layer contains entries that are strictly positive, though they can be made arbitrarily small. However, the probability distribution obtained from an $n$-gram language model typically contains many zero entries corresponding to words that cannot logically follow a given $n$-gram. The $\epsilon$ parameter is used to account for the discrepancy between the two types of language models. While $\epsilon$ can be replaced by zero if we use a hardmax or sparsemax instead of the softmax (see Section 2 of Svete & Cotterell (2024) for definitions), we opt for the softmax function as it is used more widely in empirical settings. We also note that our existence results hold for any value of the embedding dimension $e$, including $e = 1$; see the discussion after Proposition 4.5 for an explanation of why this is the case.

## 4.2 Proof: construction of the state space model.

First we introduce some additional notation to specify the algebraic rules that encapsulate the next-word prediction task. Recall that the input to a language model is a sequence of words $\underline{w}'$ that is a valid starting sequence, and the output is a vector that indicates which words can be added to the end, so that the resulting sequence can be completed to obtain a valid sentence. Note that only the last $n$ words are needed for the prediction. We formalize this below, assuming that the sequence $\underline{w}'$ lies in $\mathcal{P}$, the set of all valid $n$-grams in the language $\mathcal{L}$.

**Definition 4.2.** Given a sequence of words $\underline{w}' = \{w_1, \cdots, w_{n-1}\} \in \mathcal{P}$, recall that the vector $f_{ng}(\underline{w}')$ is a probability vector in $\Delta^d$ (recall that $d = |\mathcal{W}|$). For each $\underline{w}' \in \mathcal{P}$, we choose the vector $s_{\underline{w}'} \in \mathbb{R}^{d+1}$ with all coordinates being strictly positive with the property that $||s_{\underline{w}'} - f_{ng}(\underline{w}')||_1 < \epsilon$. We also choose a vector $\bar{s}_{\underline{w}'}$ with the property that $\text{softmax}(\bar{s}_{\underline{w}'}) = s_{\underline{w}'}$.

Above, the vector $s_{\underline{w}'}$ can be constructed by replacing all of the zero coordinates in $f_{ng}(\underline{w}')$ with strictly positive values that are smaller than $\epsilon$, and modifying the other coordinates accordingly so that the entries have sum 1, ensuring that the vector lies in $\Delta^d$. The coordinates of $\bar{s}_{\underline{w}'}$ can be obtained by taking logarithms of the entries in $s_{\underline{w}'}$, noting that the image of the softmax function is precisely those vectors in $\Delta^d$ which have strictly positive coordinates.

Our state space model processes a given input sequence $\underline{w}$ by converting it to the vector representation $E(\underline{w})$ using the embedding layer. We will construct a state space model that maps the vector $E(\underline{w})$, corresponding to a given valid starting sequence $\underline{w}'$, to the vector $\bar{s}_{\underline{w}_{k-n+1:k-1}}$ specifying the probability distribution. This will ensure that the state space language model $f_{\mathcal{S}}^*$ will be $\epsilon$-equivalent to the $n$-gram model $f_{ng}^*$, due to the bound from Definition 4.2 above. We formalize this below, noting that only $\underline{w}_{k-n+1:k-1}$, the last $n-1$ words in the input sequence, are needed to predict the next word. Note that Theorem 4.1 follows as a direct consequence of Theorem 4.3 and Proposition 4.4.

**Theorem 4.3.** *Let $\mathcal{S}$ be a one-layer state space model, such that the following holds, where $\underline{w} = (w_1, \cdots, w_{k-1})$ is any valid starting sequence, and $E$ is any embedding function.*

$$f_{\mathcal{S}}(E(\underline{w})) = \bar{s}_{\underline{w}_{k-n+1:k-1}}$$

*Let $f_{\mathcal{S}}^*$ be the state space language model obtained by concatenating the state space model $\mathcal{S}$ with an embedding layer $E$. Then $f_{\mathcal{S}}^*$ and $f_{ng}^*$ are $\epsilon$-equivalent when restricted to $\mathcal{L}$.* ∎

**Proposition 4.4.** *There exists a state space model $\mathcal{S}$ with $e$ input neurons, one hidden layer containing $|\mathcal{P}|$ neurons, and $d+1$ output neurons such that the following holds (here $\underline{w} = (w_1, \cdots, w_{k-1})$ is any valid starting sequence).*

$$f_{\mathcal{S}}(E(\underline{w})) = \bar{s}_{\underline{w}_{k-n+1:k-1}}$$

**Example.** We revisit the example from Section 3.2 to help clarify the notation used in this subsection. Let $\underline{w}$ be the valid starting sequence below, noting that $k = 10$ and $n = 4$.

$$\underline{w} = [\text{"Ron", "woke", "up", "at", "one", "o'clock", "and", "was", "too"}]$$
$$\underline{w}_{k-n+1:k-1} = [\text{"and", "was", "too"}]$$

To compute $\bar{s}_{\underline{w}_{k-n+1:k-1}}$, we start with a vector that has values $\frac{1}{4}$ for the positions corresponding to the words "upset", "angry", "elated" and "excited", and zero values elsewhere. Then $s_{\underline{w}_{k-n+1:k-1}}$ is obtained by changing all values by an amount smaller than $\epsilon$ so that all values are strictly positive, and the sum of all entries is still equal to 1. Finally, the vector $\bar{s}_{\underline{w}_{k-n+1:k-1}}$ is obtained by taking the logarithms of the coordinates of the vector $s_{\underline{w}_{k-n+1:k-1}}$, which is possible as they are strictly positive.

To prove these results, we outline the two key Propositions that are needed in Sections 4.3 and 4.4; see Appendix A for detailed proofs. Theorem 4.3 follows from the definitions in Section 3, and Proposition 4.4 follows as a direct consequence of Proposition 4.5 and Proposition 4.6.

### 4.3 Proofs: on the memorization capacity of state space models.

To construct a state space models which satisfies this property, we analyse their memorization capacity in the below proposition. This is an extension of existing results on the memorization capacity of feedforward neural networks Zhang et al. (2017), which show that they can fit an arbitrary finite set of input-output vectors (see also Huang & Huang (1991), Yun et al. (2019)). Our approach is constructive in nature, with the parameters constructed algebraically so that the equalities are satisfied. Due to the recurrence used to compute hidden states in this context, the proof is more subtle than the analogous construction for feedforward neural networks from Theorem 1 in Zhang et al. (2017). See Appendix A.1 for a detailed proof.

**Proposition 4.5.** *Let $(x_i, y_i)_{i=1}^K$ be a finite set of input-output pairs, where each input sequence $x_i = (x_{i,1}, \cdots, x_{i,n})$ consists of $n$ vectors with $x_{i,j} \in \mathbb{R}^p$, and $y_i \in \mathbb{R}^q$. Assume that the input sequences are distinct, i.e. $x_i \neq x_j$ for $i \neq j$. Then there exists a state space model $\mathcal{S}$ with a single hidden layer with $K$ neurons that can memorize this data. Specifically, there exist weight matrices $B \in \mathbb{R}^{p \times K}$, $A \in \mathbb{R}^{K \times K}$, $C \in \mathbb{R}^{K \times q}$ and bias vectors $b_y \in \mathbb{R}^q$ such that for each sequence of input $x_i$, the state space model produces the corresponding output $y_i$. Further, the state transition matrix $A$ can be chosen so that $A^n = 0$.*

The key technical step is Lemma A.2 which show that, for a generic choice of weight matrices $A$ and $B$, the hidden state vectors corresponding to the inputs are linearly independent. To prove this, first Lemma A.3 shows that for a generic choice of $A$ and $B$, the $j$-th coordinates of the hidden state vectors are pairwise distinct for each $1 \leq j \leq K$. Using this property, Lemma A.4 constructs bias terms $b_y$ such that the hidden states obtained after applying the ReLU activation are linearly independent. Using the linear independence property, we complete the proof of Proposition 4.5 by showing there exists a unique fully connected weight matrix $C$ such that the outputs of the SSM are the vectors $y_i$. In order to apply this proposition in our context, we will need to impose the additional condition that the state transition matrix $A$ is nilpotent of order $n$, and our method can be employed with this constraint. See Section 6 for a discussion of how these techniques can be used to obtain optimal bounds on the memorization capacity of state space models.

We also note that Proposition 4.5 holds for any input dimension $p$; this is used to show that Theorem 4.1 holds for any value of the embedding dimension $e$. We do not need any restrictions on $p$ in this setting, as our proof shows that Lemma A.3, which states the individual coordinates of the hidden state vectors are pairwise distinct, holds generically (even when $p = 1$). The bias term $b_y$ is then used to ensure that the resulting hidden state vectors are linearly independent. We note that our SSM model will be less expressive if the bias term is not used, and restrictions on $p$ and $e$ may be required to prove analogous results in that setting.

### 4.4 Proofs: on the context window of state space models.

Our second key result establishes how we can restrict the state space model's memory to a fixed context window by imposing restrictions on the state transition matrix. This is crucial for modeling $n$-gram languages, where the prediction of the next word only depends on the preceding $n$ tokens. The result is formalized in Proposition 4.6 below: if the state transition matrix of the state space model is nilpotent of order $n$, then the state space model's output at any time step depends only on the last $n$ input tokens. We note that restrictions on the eigenvalues of the weight matrix has been explored as means of mitigating exploding/vanishing gradients in Helfrich & Ye (2020); Kerg et al. (2019).

**Proposition 4.6.** *Let $\mathcal{S}$ be a state space model such that the state transition matrix $A$ is nilpotent of order $n$. Then given an input sequence $\underline{x} = (x_1, \cdots, x_T)$, the output $f_{\mathcal{S}}(\underline{x})$ only depends on the last $n$ input values $(x_{T-n+1}, \cdots, x_T)$.*

Our proof involves a linear algebra calculation with powers of the state transition matrix, obtained by using the convolutional view of the state space model (see Section 2.1 of Wang & Xue (2024) for more details). This formula expresses the hidden state $h_T$ at time $T$ as a sum of past inputs multiplied by a power of the state transition matrix. When the state transition matrix $A$ satisfies $A^n = 0$, the terms corresponding to inputs older than $h_{T-n}$ vanish, and $h_T$ depends only on the most recent $n$ inputs $(x_T, \cdots, x_{T-n+1})$. This shows that constraining the state transition matrix is nilpotent with order $n$ effectively bounds to the context window of the model to have length $n$. See Figure 3 below for an illustration of the proof, and Appendix A.2 for a detailed proof.

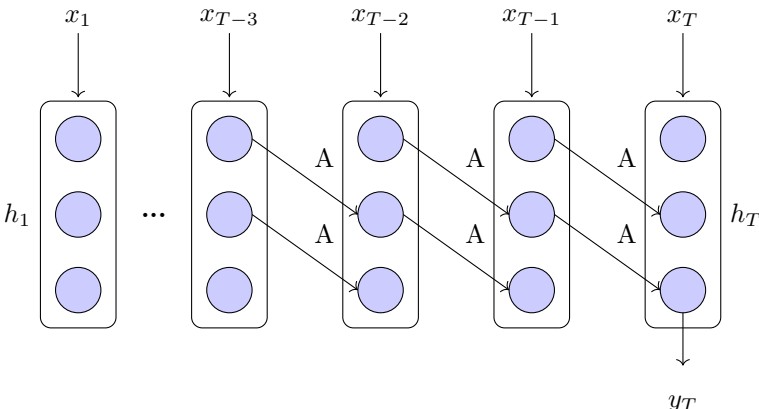

Figure 3: This figure shows how the context window state space model can be bounded using a nilpotent state transition matrix $A$. In this example, $A^3 = 0$, as shown by the arrows which represent the action of $A$ on the individual neurons, which map each neuron to a neuron with a larger index. These neurons represent the Jordan basis for the nilpotent, and the arrows indicate how all inputs prior to $x_{T-2}$ do not influence the output $y_T$.

### 4.5 Extensions to recurrent neural networks.

Recurrent neural networks (RNNs) and state space models both rely on sequential updates of a hidden state to process input sequences, with the hidden state at each time step being determined by the previous hidden state and the current input. The key difference is that the hidden state update for RNNs uses non-linearity and bias terms. Unlike the S3 model which can compute multiple time steps simultaneously due to the linear hidden state update, the output of a RNN cannot be parallelized during training Gu et al. (2021). We now outline how the key results of this section can be extended to recurrent neural networks.

Our results in Proposition 4.6 on the context window of state space layers can be extended to this setting with one key modification. In addition to the hidden weight matrix $A$ satisfying the nilpotency constraint $A^n = 0$, we also impose a condition that it is block nilpotent with respect to the standard basis formed by the hidden neurons. This is imposed to ensure that the piecewise linear map $\sigma \circ A$ satisfies the same relation; see Appendix A for a more precise definition and proof. We expect that Proposition 4.5 can also be extended to this setting; the key step is to establish the linear independence of the hidden state vectors obtained from each of the input sequences, noting that in this setting the ReLU function is applied during the hidden state update.

## 5 Experimental results: learning $n$-gram rules with state space models.

In this section, we conduct experiments with state space models trained on datasets generated synthetically from $n$-gram rules using the example in Section 3, and show that they are capable of predicting the next word with near-perfect accuracies. We study the spectral decomposition of the hidden weight matrices, and use it to analyse the context window of the state space model. We also illustrate how the hidden state vectors in the state space model encode $n$-gram rules, by visualizing the vectors with a cluster plot using principle component analysis. These results show how our theoretical framework can be used to gain insights into the inner workings of state space models trained on simple datasets using gradient-based optimization.

### 5.1 Experimental set-up

We generate data using the language $\mathcal{L}$ which is defined in Appendix B using a list of $n$-gram rules, and explained with a simplified diagram in Section 3.2. To generate a sentence from $\mathcal{L}$, see the illustration in Figure 2 (note however that this diagram does not display the full set of rules, but they are listed in the

table from Appendix B). The language $\mathcal{L}$ consists of hundreds of sentences that are grammatically correct. From the discussion in Appendix B and the example in Section 3.2, an $n$-gram language model $f_{ng}$ with $n = 4$ and 145 valid trigrams in $|\mathcal{P}|$ can be used to solve next-word prediction for this language $\mathcal{L}$. We train a state space language model for next-word prediction on this dataset. Following the setup in Section 4.3, the model consists of an embedding layer with dimension 10, a hidden layer with 145 hidden neurons, and a fully connected layer that produces the output logits. We note that the size of the hidden layer is equal to the number of valid trigrams in our model, following the setup in Theorem 4.1.

Instead of using a pre-trained word embedding (based on algorithms such as word2vec Mikolov et al. (2013a;b) or GloVe Pennington et al. (2014)), in our experiments we train the embedding layer jointly with the state space model. While our theoretical construction works with an embedding layer with arbitrary dimension $e \geq 1$, empirically we find that stochastic gradient descent does not converge when the dimension $e$ is very small. The model is trained for 25 epochs with a fixed learning rate of 0.001, using stochastic gradient descent with the cross-entropy loss function. We evaluate the model's accuracy on an unseen test set; each sentence in the test set is truncated to a length of 6, and the model's completion is evaluated by checking if the resulting sentence lies in our dataset. After replicating the experiment 5 times with different random seeds, we find the model consistently achieves near-perfect accuracies when using a training dataset with 40 sentences.

## 5.2 Spectral decomposition of hidden weight matrices

In this section, we revisit the theoretical framework in Proposition 4.6, and apply these insights to state space models that are obtained through gradient-based optimization. The key feature of our theoretical construction is that the state transition matrix is nilpotent, i.e. that it has zero eigenvalues. The below plot shows the eigenvalues derived from the state transition matrix of the recurrent language model. Each blue dot represents an eigenvalue, plotted in the complex plane with the real part on the x-axis and the imaginary part on the y-axis, while the red dashed circle indicates the unit disk.

In an empirical setting, we note that the nilpotency condition on the matrix must be relaxed; instead of stipulating that a power of $A$ is exactly 0, we check whether it is approximately equal to 0. Using the eigen-decomposition of a matrix, it is well-known that this will be true for $n$ sufficiently large precisely if all of the eigenvalues have norm less than 1. The diagram shows that the eigenvalues all lie within the unit circle, with only one exception, which supports our hypothesis.

To extend Lemma 4.6 to RNNs, as described in Appendix A, we impose the additional condition that the hidden weight matrix $A$ is block nilpotent. This implies that $(\sigma \circ A)^n$ is the zero map; note here $\sigma \circ A$ is a piecewise linear function. Unlike our observations for SSMs above, we find that for RNNs the matrix $A$ has multiple eigenvalues with values greater than 1. However, by computing the norm of vectors $(\sigma \circ A)^n E(\underline{w})$ for varying input sequences $\underline{w}$, we find that nevertheless the map $(\sigma \circ A)^n$ approaches 0 as $n$ increases. Using these observations based on the theory in Lemma 4.6, bounding the Lipschitz norm of the map $\sigma \circ A$ allows us to estimate the effective context window of an RNN (here we use Lipschitz norm instead of operator norm as this map is non-linear).

## 5.3 Hidden embeddings for $n$-grams

In this section, we revisit the theoretical framework from Proposition 4.4, and apply these insights to the state space models obtained through gradient-based optimization. The key feature of our theoretical construction there is that given an input sequence of words $\underline{w}$, the image in the hidden layer depends only on the $n$-gram obtained from the last $n$ words of the sequence; in other words, the $n$-gram rule is encoded as a vector in the hidden layer. Our findings build on an earlier study in Karpathy et al. (2015), which investigates the internal mechanisms of LSTMs and finds interpretable behaviors in individual neurons that track specific attributes, such as quotes or parenthesis.

To evaluate this hypothesis, below we apply principal component analysis (PCA) to compress the 64-dimensional hidden embedding vectors obtained from the state space model to 2-dimensional vectors that we visualize in Figure 4 above. Each point in the scatter plot represents a hidden state corresponding to a specific input sentence, with the colors of the points corresponding to the $n$-gram consisting of the last

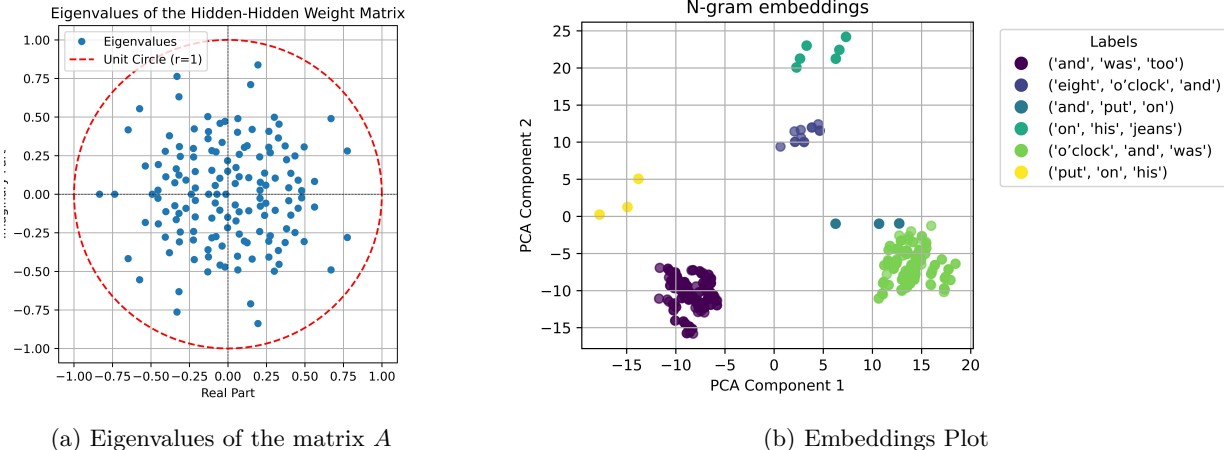

(a) Eigenvalues of the matrix $A$

(b) Embeddings Plot

Figure 4: Plot showing the eigenvalues of the matrix $A$, and hidden state embeddings corresponding to $n$-grams.

three words. This visualization illustrates the state space model's ability to encode $n$-gram rules, as vectors corresponding to the same $n$-gram are clustered together. Since input sequences ending in the $n$-gram rule have similar representations in the hidden layer, the output logits will also have similar values, enabling the state space model to predict the next word accurately.

### 5.4 Extension to transformer models

Transformers employ self-attention mechanisms, which enable them to directly attend to all tokens in a sequence, making them highly effective for long-context language modeling. It is known theoretically that transformers are more expressive than $n$-gram models Svete & Cotterell (2024), but all of the attention heads in their construction use query-key matrices that only use positional encodings and not the word embeddings. Empirically it has been observed that while some attention heads are positionally-based, most of them capture semantic information; see Section 5.1 of Eldan & Li (2023) for an analysis of transformers trained on the TinyStories dataset. In Elhage et al. (2021), by studying attention heatmaps it has been observed that transformers learn skip-gram rules, rather than $n$-gram rules (see also Nguyen (2024), where skip-gram rules are defined with the $*$ character and extracted for transformers trained on TinyStories). To address this, it would be interesting to extend our theoretical framework to analyze how transformer models encode skip-gram rules. Existing results on the memorization capacity of transformer models Mahdavi et al. (2024) can be used as a substitute for Proposition 4.5 in this context. While the nilpotency constraint from Proposition 4.6 isn't applicable to transformers, different attention heads in the transformer could encode different skip-gram rules.

## 6 Discussion: limitations and further directions.

**Tight bounds on the memorization capacity of state space models.** In our proof of Proposition 4.5, we extend the construction for feedforward neural networks with a single hidden layer from Theorem 1 of Zhang et al. (2017) to state space models. The construction uses a two-layer network with $K$ hidden nodes and $O(K^2)$ parameters to memorize $K$ input-output pairs. In Section 3 of Yun et al. (2019), it is shown that a three-layer feedforward neural network with $O(K)$ parameters can memorize $K$ input-output pairs, and that this construction is optimal (i.e. if the network had less than $K$ parameters, it cannot memorize an arbitrary set of $K$ input-output pairs). It would be interesting to extend the techniques from Yun et al. (2019) to our setting, and construct state space models with two hidden layers and $O(K)$ parameters that can memorize any set of $K$ input-output pairs. Due to the recurrence used to compute hidden states for the

state space model, the analogous construction for feedforward neural networks from Section 3 of Yun et al. (2019) does not carry over verbatim, but we expect that it can be adapted to this context.

**Generalizations of $n$-gram languages.** The present work shows that using the simplified models based on $n$-gram languages can provide insights into the workings of state space models and RNNs trained on next-word prediction tasks. However, it is well-known that $n$-gram languages cannot effectively model the English language in its entirety, as the number of $n$-gram rules required to do so will be prohibitively large Liu et al. (2024). More intricate models based on formal grammars Chomsky (1956; 1959); Chomsky & Schützenberger (1963), which have been developed by the mathematical linguistics community, are better suited for complex real-world datasets. While existing results on the Turing completeness of RNNs and transformers show the existence of models that can solve the recognition problem for formal grammars Pérez et al. (2021); Chung & Siegelmann (2021), the construction uses unbounded precision (or memory), which does not explain their success in real-world scenarios. It would be interesting to extend our construction to formal grammars and use our framework to gain further insights into the inner workings of state space models and RNNs obtained through gradient-based optimization, (see Yang et al. (2024); Bhattamishra et al. (2020); Chiang et al. (2023); Strobl et al. (2024) for analogous results on transformers).

**Generalized state space models.** While the state space models we study struggle with maintaining information over long sequences, more recent work on generalized state space models extends this by incorporating the HiPPo matrix Gu et al. (2022), which helps them efficiently represent and update the state information to capture both recent and distant past inputs. While this may not be needed for $n$-gram languages where the model only needs to consider the most recent tokens, formal grammars require the model to remember recursive relationships between different parts of a sentence and maintain consistency across nested clauses. It would be interesting to extend our theoretical framework to generalized state space models Gu et al. (2022); Gu & Dao (2024), and gain a deeper understanding of the interpretability of these models which are obtained through gradient-based optimization.

**Stochastic gradient descent and overparametrization.** Our empirical results indicate that state space models and RNNs that are trained on synthetic language data generated from $n$-gram rules can achieve near-perfect accuracies. This leads us to the question of whether we can obtain theoretical guarantees showing that stochastic gradient descent yields state space models and RNNs with near-zero generalization error in this context. There has been existing work on generalization bounds for RNNs using the PAC-Learning framework Chen et al. (2020); Zhang et al. (2018), and theoretical guarantees for the convergence of stochastic gradient descent with RNNs in the overparametrized setting Allen-Zhu et al. (2019b) (building on earlier work for feedforward neural networks Allen-Zhu et al. (2019a); Zou et al. (2020)). However, these theoretical guarantees are obtained for tasks that involve memorizing input-output pairs, and cannot be directly applied to distributions that model real-world settings, such as data generated from $n$-grams models. It would be interesting to extend their theoretical guarantees to state space models and RNNs using datasets generated from $n$-gram languages, and analyze theoretically whether or not overparametrization is required in this setting.

## 7   Conclusion.

In this paper, we use languages generated from $n$-gram rules to theoretically understand how neural networks excel at language modelling tasks. We focus on next-word prediction using state space language models, and show theoretically that they are more expressive than $n$-gram language models. In our proof, we construct a state space model that encodes the $n$-gram rules which govern the next-word prediction task, and discuss how our results can be extended to recurrent neural networks. Our key theoretical results include a rigorous analysis of the memorization capacity of state space models, and demonstrate how the context window can be regulated through the spectral decomposition of the state transition matrix. We conduct experiments with a small dataset to illustrate how our framework yields insights into the inner workings of state space models obtained through gradient-based optimization. We do not anticipate any negative societal impacts, as the present work is theoretical. Our results shed light on the theoretical underpinnings of language modelling with neural networks, and we expect that it will provide insights into the inner workings of these black-box models.

**Acknowledgments**

Vinoth Nandakumar was partially supported by an RTP scholarship at the University of Sydney. Peng Mi was partially supported by an OPPO Postgraduate Research Award in Computational Science. Tongliang Liu is partially supported by the following Australian Research Council projects: FT220100318, DP220102121, LP220100527, LP220200949, IC190100031. The authors would also like to thank the anonymous reviewers and the action editor for their detailed feedback.

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

# A   Appendix A

## A.1   On the memorization capacity of state space models

In this section, we prove Proposition 4.5. First we state the below Lemma, which is the key ingredient.

**Definition A.1.** Given an input sequence $x = (x_1, \cdots, x_n)$ to a state space model $\mathcal{S}$, recall that the hidden state sequence $(h_1, \cdots, h_n)$ with $h_i \in \mathbb{R}^K$ are computed as follows (here $1 \leq i \leq n$):

$$h_t = A h_{t-1} + B x_t$$

We define the functions $h_{\mathcal{S}}, \overline{h}_{\mathcal{S}}$ as follows, which take in an input sequence and output a hidden state vector. Here recall that $\sigma$ is the ReLU function and $b_y$ is the bias term.

$$h_{\mathcal{S}}(x) = h_n \qquad \overline{h}_{\mathcal{S}}(x) = \sigma(h_n + b_y) \qquad \blacksquare$$

**Lemma A.2.** Let $\{x_i\}_{i=1}^{K}$ be a finite set of input sequences, where each input sequence $x_i = (x_{i,1}, \cdots, x_{i,n})$ consists of $n$ vectors with $x_{i,j} \in \mathbb{R}^p$. Assume that these input sequences are distinct, i.e. $x_i \neq x_j$ for $i \neq j$. Then there exist weights for a state space model $\mathcal{S}$ with a single hidden layer with $N$ neurons with the following property: the vectors $\{\overline{h}_{\mathcal{S}}(x_i)\}_{i=1}^{K}$ are linearly independent. Further, the state transition matrix $A$ can be chosen so that $A^n = 0$.

We note that above, while the input sequences $\{x_i\}_{i=1}^{K}$ are distinct ($x_i \neq x_j$ for $i \neq j$), some of their coordinates may be equal (e.g. it is possible that the first coordinates of $x_i$ and $x_j$ are equal, provided that another pair of coordinates are distinct). In order to prove the above Lemma, we will use the following two Lemmas.

**Lemma A.3.** Given a set of input sequences $\{x_i\}_{i=1}^{K}$ as above, there exists a state space model $\mathcal{S}$ with a single hidden layer with $K$ neurons with the following property: for each $1 \leq j \leq K$, the $j$-th coordinates of the vectors $\{h_{\mathcal{S}}(x_i)\}_{i=1}^{K}$ are pairwise distinct (in other words $\phi_j(h_{\mathcal{S}}(x_i)) \neq \phi_j(h_{\mathcal{S}}(x_{i'}))$ for all $1 \leq i < i' \leq K$, where $\phi_j : \mathbb{R}^K \to \mathbb{R}$ is the function that extracts the $j$-th coordinate). Further, the state transition matrix $A$ of the state space model can be chosen so that $A^n = 0$.

*Proof.* Recall that by expanding the hidden state recurrence relation iteratively using the convolutional view of the SSM, we can express $h_{\mathcal{S}}(x_i)$ as follows.

$$h_{\mathcal{S}}(x_i) = \sum_{k=0}^{n-1} A^k B x_{i,n-k}$$

We claim that for a generic choice of the matrices $A$ and $B$, for each $j$ the $j$-th coordinates of the vectors $\{h_{\mathcal{S}}(x_i)\}_{i=1}^{K}$ are pairwise distinct. To see this, below we consider the $j$-th coordinate of the difference $h_{\mathcal{S}}(x_i) - h_{\mathcal{S}}(x_{i'})$, where $i \neq i'$; recall $\phi_j : \mathbb{R}^K \to \mathbb{R}$ is the function that extracts the $j$-th coordinate.

$$\phi_j(h_{\mathcal{S}}(x_i) - h_{\mathcal{S}}(x_i)) = \sum_{k=0}^{n-1} \phi_j(A^k B(x_{i,n-k} - x_{i',n-k}))$$

Each of the terms $\phi_j(A^k B(x_{i,n-k} - x_{i',n-k}))$ is a polynomial in the coordinates of the matrices $A$ and $B$ with degree $k+1$. Further, it is a sum of monomials, so that all the coefficients have value 1. Since $i \neq i'$, the set $S$ defined below is non-empty.

$$S = \{k \mid x_{i,n-k} - x_{i',n-k} \neq 0\}$$

It follows that for each $k \in S$, the coordinate $\phi_j(A^k B(x_{i,n-k} - x_{i',n-k}))$ is non-zero for a generic choice of matrices $A$ and $B$. It follows that their sum $\phi_j(h_{\mathcal{S}}(x_i) - h_{\mathcal{S}}(x_{i'}))$ is also non-zero for a generic choice of $A$ and $B$, since the degrees of the multivariate polynomial constituents are distinct. This statement remains true for a generic choice of nilpotent matrix $A$ satisfying $A^n = 0$. This is true because any such matrix has the property that all coordinates of $A^k$ are generically non-zero when $k < n$. $\square$

**Lemma A.4.** *Let $\{v_i\}_{i=1}^{K}$ be a set of vectors, with $v_i \in \mathbb{R}^K$, satisfying the following property: for each $1 \leq j \leq n$, the $j$-th coordinates of the vectors $\{v_i\}_{i=1}^{K}$ are pairwise distinct (in other words, $\phi_j(v_i) \neq \phi_j(v_{i'})$ for all $1 \leq i < i' \leq K$). Then there exists a vector $b_y \in \mathbb{R}^K$, such that the vectors $\{\sigma(v_i + b_y)\}_{i=1}^{K}$ are linearly independent.*

*Proof.* The vectors $\{\sigma(v_i + b_y)\}_{i=1}^{K}$ are linearly independent precisely if the determinant of the matrix $A$ obtained by concatenating these vectors is non-zero. We proceed by induction to construct the vector $b_y \in \mathbb{R}^K$. The $K = 1$ case is clear, as given any vector $v_1 \in \mathbb{R}$ we can choose a bias $b_y$ so that the determinant $\sigma(v_1 + b_y) \neq 0$. For the induction step, we assume that the statement is true for every set of $K-1$ vectors with the stated property.

Recall that the determinant of the $K \times K$ matrix $A$ can be computed using the Laplace expansion along the first row. Here $a_{ij}$ is the $(i,j)$-th entry of $A$, and where $A_{1j}$ is the $(K-1) \times (K-1)$ submatrix obtained by removing the first row and the $j$-th column of $A$.

$$\det(A) = \sum_{j=1}^{K}(-1)^{1+j}a_{1j}\det(A_{1j})$$

Note $a_{1j} = \sigma(b_{y,1} + v_{j,1})$, where $b_y = (b_{y,1}, \cdots, b_{y,K})$ and $v_j = (v_{j,1}, \cdots, v_{j,N})$ are vectors in $\mathbb{R}^N$. Since the values $v_{j,1}$ are pairwise distinct, we can choose $b_{y,1}$ so that there exists a unique value of $j$ for which $\sigma(b_{y,1} + v_{j,1}) > 0$, and $\sigma(b_{y,1} + v_{j',1}) = 0$ for all $j' \neq j$. With this choice of parameters, the determinant can be expressed as follows.

$$\det(A) = (-1)^{1+j}\sigma(b_{y,1} + v_{j,1})\det(A_{1j})$$

By using the induction hypothesis, we can choose the bias terms $(b_{y,2}, \cdots, b_{y,K})$ so that $\det(A_{1j}) \neq 0$. Here note that $\det(A_{1j})$ corresponds to the set of $K-1$ vectors in $\mathbb{R}^{K-1}$ obtained by deleting the first coordinates from the vectors $\{v_1, \cdots, v_{j-1}, v_{j+1}, \cdots, v_K\}$. It follows that for this choice of bias $b_y$, $\det(A) \neq 0$, completing the induction step. $\square$

*Proof of Lemma A.2.* First we use Lemma A.3 to choose the weight matrices $A, B$ for the state space model $\mathcal{S}$ so that, for each $j$, the $j$-th coordinates of the vectors $\{h_{\mathcal{S}}(x_i)\}_{i=1}^{K}$ are pairwise distinct ($A$ can also be chosen to be nilpotent with $A^n = 0$). Noting that $\overline{h}_{\mathcal{S}}(x_i) = \sigma(h_{\mathcal{S}}(x_i) + b_y)$, we can now use Lemma A.4 to choose the bias vectors $b_y$ so that the vectors $\{\overline{h}_{\mathcal{S}}(x_i)\}_{i=1}^{K}$ are linearly independent. $\square$

The proof of Proposition 4.5 now follows from the linear independence statement in Lemma A.2, by choosing the weights of the fully connected matrix suitably.

*Proof of Proposition 4.5.* From Lemma A.2, it follows that there exist weights such that the vectors $\{\overline{h}_{\mathcal{S}}(x_1), \cdots, \overline{h}_{\mathcal{S}}(x_K)\}$ are linearly independent. In fact, it is shown that this is true for a generic choice of weight matrices $A, B$ for the state space model $\mathcal{S}$, and that the weight matrix $A$ can also be chosen to be nilpotent with $A^n = 0$.

Using the linear independence, it is now clear that there exists a weight matrix $C$ such that the following holds for each $i$ with $1 \leq i \leq K$, completing the proof.

$$y_i = C\overline{h}_{\mathcal{S}}(x_i) = f_{\mathcal{S}}(x_i)$$

$\square$

## A.2 On the context window of state space models and recurrent neural networks.

We first describe the proof of Proposition 4.6, which follows from using the convolutional view of state space models.

*Proof of Proposition 4.6.* Recall that the sequence of outputs produced is denoted $(y_1, \cdots, y_T)$, and the sequence of intermediary hidden states as $(h_1, \cdots, h_T)$. By expanding the hidden state recurrence relation iteratively, we can express $h_t$ in terms of previous inputs $x_t, x_{t-1}, \ldots, x_1$ as follows:

$$h_T = \sum_{k=0}^{n} A^k B x_{T-k}$$

When $k \geq n$, note that $A^k = 0$; it follows that the value of $h_T$ depends only on the values $(x_T, \cdots, x_{T-n+1})$ (and not $x_{T-n}$ or any earlier inputs). $\qquad\square$

## A.3 Proof of the main results for state space models.

Theorem 4.1 follows immediately as a consequence of Theorem 4.3 and Proposition 4.4. We now describe the proof of Theorem 4.3 and Proposition 4.4.

*Proof of Proposition 4.4.* We choose a SSM whose state transition matrix $A$ is nilpotent with order $n-1$. By the above Lemma, $f_{\mathcal{S}}(E(\underline{w}))$ only depends on the last $n-1$ words of $\underline{w}$; we denote the result by $f_{\mathcal{S}}(E(w_{k-n+1}, \cdots, w_{k-1}))$. To complete the proof, we need to choose the weights of $\mathcal{S}$ so that the following holds.

$$f_{\mathcal{S}}(E(w_{k-n+1}, \cdots, w_{k-1})) = \overline{s}_{\underline{w}_{k-n+1:k-1}}$$

We can do this using the above Lemma on memorizing data with SSMs, by setting $x_i = f_{\mathcal{S}}(E(w_{k-n+1}, \cdots, w_{k-1}))$ and $y_i = \overline{s}_{\underline{w}_{k-n+1:k-1}}$. $\qquad\square$

*Proof for Theorem 4.3.* We show that $f_{\mathcal{S}}^*$ and $f_{ng}^*$ are $\epsilon$-equivalent when restricted to $\mathcal{L}$ below by unravelling the definitions from Section 3. It suffices to prove that for any valid starting sequence $\underline{w} \in \mathcal{L}_k$, for some $k$, the following holds.

$$||f_{\mathcal{S}}^*(\underline{w}) - f_{ng}^*(\underline{w})||_1 < \epsilon$$

This follows from the below chain of equalities.

$$
\begin{aligned}
f_{\mathcal{S}}^*(\underline{w}) &= \mathrm{softmax}(f_{\mathcal{S}}(E(\underline{w})) \\
&= \mathrm{softmax}(\overline{s}_{\underline{w}_{k-n+1:k-1}}) \\
&= s_{\underline{w}_{k-n+1:k-1}} \\
f_{ng}^*(\underline{w}) &= f_{ng}(\underline{w}_{k-n+1:k-1}) \\
||f_{\mathcal{S}}^*(\underline{w}) - f_{ng}^*(\underline{w})||_1 &= ||\overline{s}_{\underline{w}_{k-n+1:k-1}} - f_{ng}(\underline{w}_{k-n+1:k-1})||_1 \\
&< \epsilon
\end{aligned}
$$

$\qquad\square$

## A.4 Extensions to RNNs.

First we recall the definition of a recurrent neural network. While the hidden weight matrices are often denoted as $W_x, W_h, W_y$ in the literature, for consistency we use the notation $A, B, C$ from Section 3.

**Definition A.5.** A *one-layer recurrent neural network* (RNN) $\mathcal{R}$ consists of an input layer with $n_0$ input neurons, a hidden layer with $n_h$ hidden neurons, and an output layer with $n_L$ output neurons. The weight matrices are denoted by $A$, $B$, and $C$, where $A \in \mathrm{Mat}_{n_h, n_h}(\mathbb{R})$ is the hidden-to-hidden weight matrix, $B \in \mathrm{Mat}_{n_0, n_h}(\mathbb{R})$ is the input-to-hidden weight matrix, and $C \in \mathrm{Mat}_{n_h, n_L}(\mathbb{R})$ is the hidden-to-output weight matrix. Further, $b_h \in \mathbb{R}^{n_h}$ and $b_y \in \mathbb{R}^{n_L}$ are the bias vectors for the hidden and output layers, respectively.

It yields a function $f_{\mathcal{R}} : (\mathbb{R}^{n_0})^T \to (\mathbb{R}^{n_L})^T$; the RNN processes a sequence of inputs $(x_1, x_2, \ldots, x_T)$, produces a sequence of hidden states $(h_1, h_2, \ldots, h_T)$, and yields outputs $(y_1, y_2, \ldots, y_T)$ via the following equations. Here $1 \leq t \leq T$.

$$h_t = \sigma(Bx_t + Ah_{t-1} + b_h) \qquad y_t = Ch_t + b_y \qquad \blacksquare$$

Now we show how Proposition 4.6 can be extended to RNNs, by imposing the block nilpotency condition on the hidden weight matrix $A$.

**Definition A.6.** A matrix $M = (M_{ij})_{1 \le i \le k, 1 \le j \le k} \in \mathrm{Mat}_k(\mathbb{R})$ is *block nilpotent of order $n$* if $M^n = 0$ and the induced map $\phi_M : \mathbb{R}^k \to \mathbb{R}^k$ satisfies the following property: for each $i$, the space $\ker(\phi_M^i)$ has a basis consisting of standard vectors in $\mathbb{R}^k$ (here a standard vector is a vector in $\mathbb{R}^k$ with exactly one non-zero coordinate, with value 1). ∎

**Lemma A.7.** *Let $\mathcal{R}$ be a recurrent neural network such that the hidden weight matrix $A$ is block nilpotent of order $n$. Then given an input sequence $\underline{x} = (x_1, \cdots, x_T)$, the output $f_{\mathcal{R}}(\underline{x})$ only depends on the last $n$ input values $(x_{T-n+1}, \cdots, x_T)$.*

*Proof of Lemma A.7.* Denote the sequence of outputs as $(y_1, \cdots, y_T)$, and the sequence of intermediary hidden states as $(h_1, \cdots, h_T)$. Consider the kernel $\ker(A)$; it is easy to see that $h_T|_{\ker(A)}$ depends only on the value of $x_T$ (and not $x_{T-1}$ or any earlier inputs). It follows that $h_T|_{\ker(A^2)}$ depends only on the value of $(x_T, x_{T-1})$ (and not $x_{T-2}$ or any earlier inputs). By induction, we see that $h_T|_{\ker(A^i)}$ depends only on the value of $(x_T, \cdots, x_{T-i+1})$ (and not $x_{T-i}$ or any earlier inputs). When $i = n$, note that $A^n = 0$ and $\ker(A^n) = \mathbb{R}^{n_h}$; it follows that the value of $h_T$ and $y_T$ depend only on the values $(x_T, \cdots, x_{T-n+1})$ (and not $x_{T-n}$ or any earlier inputs). □

We now discuss how Theorem 4.1 can be extended to RNNs. The proofs in Section A.3 can be applied in this setting without any changes once Proposition 4.6 and Proposition 4.5 have been established. The only remaining ingredient is Proposition 4.5, and we now describe how the proofs in Section A.1 can be extended to RNNs. To show that Proposition 4.5 holds for RNNs, we need to show that the linear independence statement in Lemma A.2 holds in this context; the proof of Proposition 4.5 in Section A.1 can then be applied without any changes. To show Lemma A.2, note that the bias term $b_h$ and the non-linearity both appear in the hidden state update. However, when the bias term $b_h$ has entries which are sufficiently large, the non-linearities are redundant as all the values are positive. The proof of Lemma A.3 can then be used to establish the linear independence, by viewing the coordinates as polynomials in the bias $b_h$ and the weight matrices $A$ and $B$.

### A.5 Bounding the memorization capacity of state space models.

In this section, we revisit Proposition 4.5 and investigate the following question: what is the minimum size of a state space model $\mathcal{S}$ with the property that it can memorize an arbitrary set of $K$ input-output pairs? Following the setup in Theorem 3.3 of Yun et al. (2019), for simplicity we assume that the outputs $y_i \in \mathbb{R}$ (i.e. $q = 1$). Below we extend the statement in that Theorem for two-layer feedforward neural networks to state space models, which shows that Proposition 4.5 is tight in the setting where $q = 1$.

**Proposition A.8.** *Suppose $\mathcal{S}$ is a state space model with $p$ input neurons, a single hidden layer with $k$ neurons, and one output neuron. Suppose that $k < N$. Then there exists a set of input-output pairs $(x_i, y_i)_{i=1}^N$ with $x_i \in \mathbb{R}^p, y_i \in \mathbb{R}$ with the following property: for any choice of weights matrices $A, B, C$ for $S$, there exists $i$ such that $f_{\mathcal{S}}(x_i) \ne y_i$.*

*Proof.* Let $u \in \mathbb{R}^p$ be any non-zero vector. Following Yun et al. (2019), we choose $x_i = iu$ and $y_i = (-1)^i$. We define the piecewise linear function $g(t) = f_{\mathcal{S}}(tu)$, and assume for sake of contradiction that $f_{\mathcal{S}}(x_i) = y_i$ for all $1 \le i \le N$. It follows that $g(i) = (-1)^i$ for $1 \le i \le N$, and hence that the function $g(t)$ has at least $N$ pieces.

Recall Lemma D.1 from Yun et al. (2019), which states if $f_1$ (resp. $f_2$) are piecewise linear functions from $\mathbb{R}$ to $\mathbb{R}$ with at most $p_1$ (resp. $p_2$) pieces, then $f_1 + f_2$ is a piecewise linear function with at most $p_1 + p_2 - 1$ pieces. It follows that by induction that if $g_1, \cdots, g_n$ are piecewise linear functions with 2 pieces, then $g_1 + \cdots + g_n$ is a piecewise linear function with at most $n + 1$ pieces.

Recall that $h_{\mathcal{S}}(x_i)$ can be expressed as follows.

$$h_{\mathcal{S}}(x_i) = \sum_{k=0}^{n-1} A^k B x_{i,n-k}$$

It follows that the functions $g(t)$ can be expressed as a sum $g(t) = g_1(t) + \cdots + g_k(t)$, where each $g_i(t)$ is a composition of the ReLu function $\sigma$ and a linear function. Since each function $g_i(t)$ has 2 pieces, it follows from the above that the piecewise linear function $g(t)$ has at most $k + 1$ pieces. This contradicts the prior deduction that $g(t)$ has at least $N$ pieces, since $k < N$. $\qquad\square$

## B  Appendix B

In this section, we elaborate on the example from Section 3.2, which is also used in the experiments in Section 5. We define a language $\mathcal{L}$ using the two sentence templates below. Sentences in $\mathcal{L}$ are generated by substituting specific words into the placeholders $(A, B, ..., I, J)$ as specified below.

- [**A**, woke, at, **B**, o'clock, and, was, too, **C**, to, go, back, to, **D**]

- [He, **E**, up, and, **F**, on his, **G**, because, he, didn't, want, to, **H**, into the, **I**, in, his, **J**]

- **A:** [Harry, Ron, Sirius, Hagrid, Fred, George, Neville, Draco, Albus, Snape]

- **B:** [one, two, three, four, five, six, seven, eight, nine, ten, eleven, twelve]

- **C:** [excited, angry, elated, agitated, nervous, troubled, upset]

- **D:** [sleep, bed]

- **E:** [got, stood, rose]

- **F:** [put, pulled]

- **G:** [jeans, trousers, pants, denims, slacks]

- **H:** [walk, run, jog, stroll, sprint, saunter, amble]

- **I:** [station, platform, house, building, apartment]

- **J:** [robes, pyjamas, shorts, underpants, boxers, trunks]

Below we list 4 sample sentences that can be generated from $\mathcal{L}$.

1. Harry woke at one o'clock and was too excited to go back to sleep.

2. Ron woke at seven o'clock and was too troubled to go back to bed.

3. He got up and put on his jeans because he didn't want to jog into the station in his robes.

4. He stood up and pulled on his trousers because he didn't want to sprint into the platform in his pyjamas.

The language $\mathcal{L}$ can be generated from an $n$-gram language model $f_{ng}^*$, with $n = 4$. The table below has a complete list of all $n$-grams in $\mathcal{P}$; note that $|\mathcal{P}| = 145$. See Section 3 for an example showing how the function $f_{ng} : \mathcal{P} \to \Delta^d$ is defined.

Table 1: All valid trigrams in $\mathcal{P}$ from the language $\mathcal{L}$.

| | | | | |
|---|---|---|---|---|
| Harry woke at | Ron woke at | Sirius woke at | Hagrid woke at | Fred woke at |
| George woke at | Neville woke at | Draco woke at | Albus woke at | Snape woke at |
| woke at one | woke at two | woke at three | woke at four | woke at five |
| woke at six | woke at seven | woke at eight | woke at nine | woke at ten |
| woke at eleven | woke at twelve | at one o'clock | at two o'clock | at three o'clock |
| at four o'clock | at five o'clock | at six o'clock | at seven o'clock | at eight o'clock |
| at nine o'clock | at ten o'clock | at eleven o'clock | at twelve o'clock | one o'clock and |
| two o'clock and | three o'clock and | four o'clock and | five o'clock and | six o'clock and |
| seven o'clock and | eight o'clock and | nine o'clock and | ten o'clock and | eleven o'clock and |
| twelve o'clock and | o'clock and was | and was too | was too excited | was too angry |
| was too elated | was too agitated | was too nervous | was too troubled | was too upset |
| too excited to | too angry to | too elated to | too agitated to | too nervous to |
| too troubled to | too upset to | excited to go | angry to go | elated to go |
| agitated to go | nervous to go | troubled to go | upset to go | to go back |
| go back to | back to sleep | back to bed | He got up | He stood up |
| He rose up | got up and | stood up and | rose up and | up and put |
| up and pulled | and put on | and pulled on | put on his | pulled on his |
| on his jeans | on his trousers | on his pants | on his denims | on his slacks |
| his jeans because | his trousers because | his pants because | his denims because | his slacks because |
| jeans because he | trousers because he | pants because he | denims because he | slacks because he |
| because he didn't | he didn't want | didn't want to | want to walk | want to run |
| want to jog | want to stroll | want to sprint | want to saunter | want to amble |
| to walk into | to run into | to jog into | to stroll into | to sprint into |
| to saunter into | to amble into | walk into the | run into the | jog into the |
| stroll into the | sprint into the | saunter into the | amble into the | into the station |
| into the platform | into the house | into the building | into the apartment | the station in |
| the platform in | the house in | the building in | the apartment in | station in his |
| platform in his | house in his | building in his | apartment in his | in his robes |
| in his pyjamas | in his shorts | in his underpants | in his boxers | in his trunks |

