# OpenReview forum: "State space models can express $n$-gram languages"
_TMLR — Accepted by TMLR_

### Review · Reviewer_FqGM · 2024-12-16

**Summary Of Contributions:**

The paper shows that SSMs have more representation power than n-gram models.

**Audience:**

Yes

**Broader Impact Concerns:**

None.

**Claims And Evidence:**

No

**Requested Changes:**

Rewrite the paper.

**Strengths And Weaknesses:**

Good: The main results are clear and bring more intuition on what SSM can do.

Bad: I do not like the paper writing.

There is so much state wasted on filler, e.g. whole example of n-gram model could be shorter, not use full words from Harry Potter, but just symbols A B C D; Section 4 is full of text, but there is no proof, all is deferred to the Appendix (more on this later).

There is no simple figure explaining the proof idea. I am not 100% sure about SSM, but I can explain a similar result for RNN with ReLU nonlinearity and one-hot encoded inputs with a simple picture (the hidden state will one-hot encode which n-gram we see, ...). Authors should explain the main proof idea in the main body of the paper (including some pictures!) and refer to technicalities in the appendix.

The experiment section writing is also not great. It refers to nonexistent Lemma 5.6. Also, it never states the size of the n-gram language (the number of valid n-grams), so it is hard to compare the number of hidden neurons (64) to the size of the language.

---

> ### Author Response · Authors · 2025-01-11
> **We have added diagrams explaining the key proofs, and added paragraphs clearly explaining the proof idea.**
>
> We thank the reviewer for their feedback, which has helped improve our manuscript. We have updated the manuscript after addressing the points that were raised; sentences that were changed are marked in blue. Please see below for more details.
>
> ""There is so much state wasted on filler, e.g. whole example of n-gram model could be shorter, not use full words from Harry Potter, but just symbols A B C D;""
> - The example of the n-gram model in Section 3.2 has been shortened substantially, and a diagram has been used to clearly illustrate how sentences in this language are formed. We use a mixture of words and symbols (A, B, C, D) to emphasize that the sentences formed are grammatically correct, and that $n$-gram models can be used to model English sentences. The table consisting of a subset of all n-gram rules has been removed from Section 3.2, and instead we refer the reader to the full table in Appendix B. The definition of a $n$-gram language has been updated using a probabilistic formulation (see Definition 3.5), and the example in Section 3.2 has also been updated to clarify the notion used there.
>
> ""There is no simple figure explaining the proof idea. I am not 100% sure about SSM, but I can explain a similar result for RNN with ReLU nonlinearity and one-hot encoded inputs with a simple picture (the hidden state will one-hot encode which n-gram we see, ...).""
> - We have added a simple figure on page 2 explaining the proof technique, with a detailed caption. We note that the hidden state vectors are not a one-hot encoding, but distinct n-gram rules correspond to distinct hidden state vectors that are linearly independent. However, the output logits essentially correspond to a multi-hot encoding, with all significant values corresponding to the words that can appear next in the sequence in accordance with the n-gram rules (all other words will have values close to 0, as explained in Sections 4.2 and 4.3). We have also added a figure in Section 4.4 explaining the proof of Proposition 4.6, illustrating how the nilpotency constraint on the state transition matrix effectively bounds the context window of the state space model.
>
> ""Authors should explain the main proof idea in the main body of the paper (including some pictures!) and refer to technicalities in the appendix. Section 4 is full of text, but there is no proof, all is deferred to the Appendix (more on this later).""
> - We have added a paragraph at the end of Section 4.3 explaining the proof of Proposition 4.5 on the memorization capacity of state space models. We have also added a paragraph at the end of Section 4.4 explaining the proof of Proposition 4.6 which bounds the context window of the state space model. We have included a simple diagram explaining this proof, illustrating how the nilpotent state transition matrix acts on the neurons in the hidden layer. We have also added a few sentences in Section 4 about the proof of Theorem 4.1, Theorem 4.3 and Proposition 4.4.
>
> ""The experiment section writing is also not great. It refers to nonexistent Lemma 5.6. Also, it never states the size of the n-gram language (the number of valid n-grams), so it is hard to compare the number of hidden neurons (64) to the size of the language.""
> - We have added details in Section 5.1, stating that the size of the n-gram language is 145. The experiments have been updated, so that the number of neurons in the hidden layer of the state space model is also 145, which is consistent with the formulation of our key Theorem. We have corrected the reference to Lemma 5.6, and it now reads Proposition 4.6. We have also added a discussion about the optimality of our construction in Section 6, and indicate how a similar result could be proven with state space models that has two hidden layers and fewer neurons by building on analogous results for feedforward neural networks.

---

> ### Author Response · Authors · 2025-01-14
>
> Dear Reviewer FqGM,
>
> We sincerely appreciate your valuable feedback, which has helped us improve the quality of our manuscript. As there are only a few days left in the discussion period, we wanted to check if you have any remaining concerns. If so, we would be glad to have further discussions.
>
> Thank you once again for your time and feedback. We look forward to hearing from you.

---

> > ### Comment · Reviewer_FqGM · 2025-01-14
> >
> > Thank you for the changes, I like the paper much more now.
> >
> > However, Figure 1 should either be removed or provide an example of internal state representation because, in its current state, it provides zero value (it is basically an overview of RNN).

---

> ### Author Response · Authors · 2025-01-16
> **Updates to Figure 1**
>
> We thank the reviewer for their quick response. We have updated Figure 1; please see below for more details. Please let us know if you have any remaining concerns for us to address. We look forward to hearing from you and thank you again for your time and feedback.
>
> """However, Figure 1 should either be removed or provide an example of internal state representation because, in its current state, it provides zero value (it is basically an overview of RNN)."""
> - We have updated Figure 1 with a cluster plot of these vectors, so that it clearly illustrates how the internal state representations correspond to the $n$-grams. These hidden state embedding vectors have been projected to two-dimensional space, and the vectors corresponding to the same $n$-gram form clusters. See also Figure 4 for a more detailed cluster plot of the internal state representations.

---

### Review · Reviewer_bNnk · 2024-12-28

**Summary Of Contributions:**

The authors study expressivity of simple linear RNNs (perhaps trained like SSMs) followed by a nonlinearity. In particular, the authors are concerned with showing SSMs, of increasing size, can represent all rules in an n-gram language. The authors start with a literature review, comprising both an overview of why SSMs are of interest and a summary of related works on n-grams representations. The authors proceed with formal definition of embeddings, n-grams (with examples), and linear recurrences. Section 4 is dedicated to the proof while Section 5 complements with empirical findings.
The heart of the paper is section 4, outlining the main result and few key propositions proved in the appendix. Essentially, the authors prove that 1 layer model SSM (without MLP but with final nonlinearity) can represent a number of n-gram rules equal to its hidden neurons.

**Audience:**

Yes

**Broader Impact Concerns:**

No concern.

**Claims And Evidence:**

Yes

**Requested Changes:**

**Changes to the proof workflow:**

I suggest making the inner workings of the proof a bit more evident in the main paper.
- Thm 4.1: it is not clear what role epsilon has. It seems it has very little effect, but then one might ask why its there! You should comment on this and provide an example. This is also related to Definition 4.2, that  is indeed very complicated and perhaps an example can help the reader understand how exactly you feed in input-output sequences.
- Thm 4.3 and Prop 4.4: here notation is very hard to parse. please put in examples.
- Relation between Prop 4.5 and prop 4.6: I feel like this is the heart of the proof but this is indeed hidden in the appendix: Prop 4.5 says that there EXISTS (A,B,C) such that... but in prop 4.6 you need A (called W_hh, please adapt) to be nihlpotent. It is unclear to me if you can use 4.5 while setting A to be nihl potent.. digging in the appendix, I found (proof of 4.5) "this is true in fact for any A,B". What is it then? does there exist a pair $(A,B)$ or can you work with nihlpotent?
- "pairwise distinct": please move definition from appendix to main paper. Why is this assumption not restrictive? does this impliy something on the possible n-gram rules you can model?
- Lemma A2 in app: $N$ neurons... do you mean $K$ neurons? same holds for $\{h_S(x_i)\}_{i=1}^N$, which I think should be $K$.

**Changes to discussion and experiments**

Please provide a comparison (empirical and theoretical) with attention-based models.

**Typos**
Please update the way to build the main paper: compile all and cut pages, so you avoid the "??" for appendix references.

--------------------------

Comments mostly solved by the authors

**Strengths And Weaknesses:**

I like very much the topic of the paper and I think it could be of great interest of the community.

Strengths:
- Topic is of great interest for the community: understanding the power of simple linear recurrences is an unexplored and important topic.
- The paper is relatively easy to read, proof are quite simple
- The authors provide a nice introduction to the topic and give all necessary background for people with little background to understand

Weaknesses:
- Proof is a bit fragmented, and some key information is hidden in short appendix comments. See requested changes section.
- The main content of the paper is not much: the proof can be condensed in a few key ideas and experiments are quite simple.
- Notation is a bit heavy at times.
- Role of some key quantities, such as the epsilon in the main theorem, is not clear and not commented.
- The paper refers to Jelassi et al. 2024 as a similar contribution: while I agree, what is missing here is a detailed comparison with transformer architectures.
- Notation can change in the paper: recurrent matrix is sometimes called A, sometimes W_hh, sometimes W_h.

---

> ### Author Response · Authors · 2025-01-11
> **We have added several sentences clarifying the points raised, fixed the notational issues, and included a more detailed comparison with attention-based models.**
>
> We thank the reviewer for their detailed and actionable feedback, which has helped improve our manuscript. We have updated the manuscript after addressing the points that were raised; sentences that were changed are marked in blue. Please see below for more details.
>
> ""Thm 4.1: it is not clear what role epsilon has. It seems it has very little effect, but then one might ask why its there! You should comment on this and provide an example.""
> - We have added a paragraph at the end of Section 4.1 explaining that the epsilon appears due to the presence of a softmax function, which yields entries that are strictly positive (but can be arbitrarily small). However, the probability distribution obtained from an $n$-gram language model typically contains many zero entries, so the $\epsilon$ parameter is used to account for this discrepancy. While $\epsilon$ can be replaced by zero if we use a hardmax or sparsemax instead, we opt for the softmax function as it is used more widely in empirical settings. This is further clarified in the Example in Section 4.2.
>
> ""This is also related to Definition 4.2,...Thm 4.3 and Prop 4.4: here notation is very hard to parse. please put in examples.""
> - We have added a detailed example in Section 4.2, following on from the example of an n-gram language model from Section 3.2. In particular, we included an example of a sequence $\underline{w}$, and calculated the corresponding set $S$ defined in Definition 4.2, which is used to define the vector referenced in Theorem 4.3 and Proposition 4.4.
>
> ""Relation between Prop 4.5 and prop 4.6:...: Prop 4.5 says that there EXISTS (A,B,C) such that... but in prop 4.6 you need A to be nilpotent. It is unclear to me if you can use 4.5 while setting A to be nilpotent.""
> - We have updated Prop 4.5, adding that the state transition matrix A can be chosen to be nilpotent with $A^n = 0$. This is required for the proof of Proposition 4.4. In the Appendix we describe why the techniques we use in the proof of Proposition 4.5 can be used to handle this additional constraint. The key ingredient is Lemma A.3, which holds for a generic choice of matrices A and B; we show that it also remains valid for a generic choice of nilpotent matrix A.
>
> "" "pairwise distinct": please move definition from appendix to main paper. Why is this assumption not restrictive? ""
> - We have updated Proposition 4.5 and Lemma A.2, replacing the phrase “pairwise distinct” with “distinct”, and defining the notion precisely. We have updated Lemma A.3 and Lemma A.4, which states that the coordinates of certain vectors are pairwise distinct, and defined this notion precisely. This should clarify any confusion regarding the terminology. The assumption in Proposition 4.5 and Lemma A.2 is not restrictive, as it simply states the input vectors are distinct from one another. In other words, this means that no two of the $n$-grams in Definition 3.4 are identical, which is a standard assumption in this context.
>
> "" Please provide a comparison (empirical and theoretical) with attention-based models. ""
> - We have added a paragraph in Section 5.4 with a detailed comparison. We cite [1], which shows theoretically that transformers are at least as expressive than $n$-gram models. We highlight that the limitation that all of the attention heads in their construction uses query-key matrices that only encode positional information and not semantic information, which is not true for transformers trained on datasets such as TinyStories (see Section 5.1 of [2]). Empirically, it has also been observed in [3] that transformers learn skip-gram rules, rather than $n$-gram rules. We discuss how our framework could yield generalizations of [1] that address these limitations using existing results on the memorization capacity of transformers.
>
> "" Typos: Please update the way to build the main paper: compile all and cut pages, so you avoid the "??" for appendix references. Lemma A2 in app: N neurons... do you mean K neurons? ... Notation can change in the paper: the recurrent matrix is sometimes called $A$, sometimes $W_{hh}$, sometimes $W_{h}$. ""
> - We have fixed the formatting issues, and all references to the Appendix are displayed correctly now. We have fixed the statement of Lemma A2, so that there are K input vectors and K neurons in the hidden layer of the state space model (instead of N).
> We have updated the notation, so that the state transition matrix for the SSM is always called $A$. In Appendix A.4, we also use $A$ for the hidden weight matrix of RNNs for consistency, noting that it is also often denoted $W_{h}$ in the literature.
>
> [1] Anej Svete, Ryan Cotterell. Transformers Can Represent n-gram Language Models, NAACL 2024
> [2] Ronen Eldan and Yuanzhi Li. Tinystories: How small can language models be and still speak coherent english? arXiv preprint
> [3] Nelson Elhage, Neel Nanda, et al. A mathematical framework for transformer circuits. 2021. Transformer Circuits Thread.

---

> ### Author Response · Authors · 2025-01-14
>
> Dear Reviewer bNnk,
>
> We sincerely appreciate your valuable feedback, which has helped us improve the quality of our manuscript. As there are only a few days left in the discussion period, we wanted to check if you have any remaining concerns. If so, we would be glad to have further discussions.
>
> Thank you once again for your time and insights. We look forward to hearing from you.

---

> > ### Comment · Reviewer_bNnk · 2025-01-18
> > **Thanks!**
> >
> > Thanks! I think the paper is in a way better shape now. Of course, it would greatly profit from empirical evidence.

---

### Review · Reviewer_4zFd · 2025-01-05

**Summary Of Contributions:**

The authors show that state space models can model an n-gram language in a certain (unconventional) sense.

**Audience:**

No

**Claims And Evidence:**

Yes

**Requested Changes:**

**Technical revisions:**
Please address or revise the technical concerns mentioned above.

I marked the Audience criteria below as a "No" based on my concern #1 above. The notion of risk is very unusual and there is no attempt to model n-gram models as probabilistic models. In my view, this needs to be addressed in order to revise the answer to "Yes".

**Notational comments:**
Many quantities are overlined and underlined and it is unclear (and confusing) what such modifications indicate (if anything). Please fix. *Examples:*

A language is denoted by an $\bar{L}$, but there is no $L$. Likewise with $E$ for embedding.

What is the difference between $f$ and $\bar{f}$?

A nonlinearity is denoted by $\sigma$ but a overline $\sigma$ is softmax.

In Definition 3.3, the notation $f(w | \underline{w})$ is used; $w$ is meant to be a dummy variable while $\underline{w}$ is fixed, yet this conflicts with $\underline{w}$ being used to denote that $w$ is a vector elsewhere.

**Expository comments:**

The example after Definition 3.1 could just be explained more straightforwardly and less confusingly. Using the words "intuitively", showing an incomplete table of n-grams, and using a partial example of substitutions like "Ron" and then finally just showing the full table in the Appendix, is wasting the reader's time when all that's needed is just to say refer to full table in the Appendix for the full list of n-gram generators.  In fact it's unclear why these were the chosen valid n-gram generators (is it just all valid n-grams from two sentences of Harry Potter as well as some formulaic word substitutions?). If so, that can just be explained without writing a very long table.

**Strengths And Weaknesses:**

**Strengths:**
1. The paper provides explicit mathematical constructions for what it aims to achieve.
2. The paper cites the literature well.

**Weaknesses:**
1. The definitions in the paper are very questionable if not incorrect. For instance, the definition of an n-gram language is just as a list of valid strings. Why is there no underlying probabilistic model (as would be the case for the usual definition for an n-gram model in which p(w | context) is given)?

    Likewise, the definition of risk is quite odd. It is equal to the average probability mass that f assigns to strings not in the n-gram language (thus zero risk means placing mass only on the supported n-gram strings of the n-gram language). But this is rather odd because it does not take into account what the distribution of f looks like on the n-gram language. What one really wants is an a priori distribution on strings and for the risk to measure the deviation from that distribution (e.g. a cross entropy type loss).

     As a degenerate example, consider a language that consists of all possible strings. Then any state space model, however initialized, has zero risk.

    Perhaps the authors had more in mind something like the recognition problem for languages (following [3]), but are restricting to n-gram languages. Then the terminology should reflect that (n-gram models are probabilistic and so are not the appropriate terminology). However I would argue this is not the correct perspective to take, since people are interested in n-gram models not n-gram languages.

2. Assuming #1 is fixed, the title appears quite suboptimal. A more accurate and direct title would be "State-space models can express
n-gram models". It's unclear why there is a question in title when the paper is putting forth a theorem: the title should just reflect that.

3. There is no discussion about the sharpness of the results. Given that, morally speaking, a sufficiently large neural network can model any finite dataset, it's not surprising that some configuration of the state-space model can represent an n-gram model. Moreover, this is more obvious for a state-space model given its linear recurrence, than for a transformer: an n-gram model can be thought of as an (n-1)-Markov model and the latter can be encoded in a sufficiently large matrix; a state-space model which "implements" this Markov transition matrix would recover the n-gram language. However, such a transition matrix would in general have as its dimensions $V^{n-1} \times V$ (where $V$ is the vocab size), i.e. the space of all possible transitions from (n-1)-grams to 1-grams. While it is interesting that the author's main theorem allows of any embedding dimension greater than or equal to 1 (this seems surprising, I have not looked carefully as to why embedding dimension = 1 works, the author should emphasis this wide range), the author's main theorem also emphasizes that their hidden state has size the number of n-grams but not the number of sentences in the language. This however isn't surprising given that the corresponding Markov matrix also has size the number of n-grams.

    If the authors could provide some analysis on (sub)optimality, the paper would be more complete in its analysis. For instance, in the cited reference [1], there are sharpness results showing that a certain capacity is needed for their representation results. It would be meaningful if the authors could do a similar analysis.

4. Summarizing the above, the authors would benefit from referring to [2], which although was cited in their work, did not seem to inform the authors as much as it should have. This is because [2] treats n-grams probabilistically. While [2] also needs hidden dimension that grows as |V|^n (again to encode the Markov matrix associated to an n-gram model), their construction at least discusses this dependence (and how it might be mitigated) and provides a variety of different constructions (varying number of heads/layers). The current paper seems to give a single construction without much context as to how (un)canonical or (not) close to optimal it may be.


[1] Chulhee Yun, Suvrit Sra, and Ali Jadbabaie. Small relu networks are powerful memorizers: a tight analysis
of memorization capacity.
[2] Anej Svete Ryan Cotterell. Transformers Can Represent n-gram Language Models
[3] Perez et al. Attention is Turing complete.

---

> ### Author Response · Authors · 2025-01-11
> **We have updated the manuscript using a probabilistic definition of $n$-gram language models, and added a detailed discussion about the optimality of our results on the memorization capacity of SSMs.**
>
> We thank the reviewer for their detailed feedback. We have updated the manuscript; sentences that were changed are marked in blue.
>
> ""Why is there no underlying probabilistic model?... What one really wants is an a priori distribution on strings and for the risk to measure the deviation from that distribution (e.g. a cross entropy type loss).""
> - We have updated Section 3.2 so that the definition of an n-gram language uses a probabilistic model following [2]. We have also rephrased the key results in Section 4 using a notion of “$\epsilon$-equivalence” between two language models (the definition of risk has been removed), which is introduced in Definition 3.5. This notion is based on the notion of “weak equivalence” from [2], and means the total variation distance between the probability distributions is less than $\epsilon$. Initially we studied the recognition problem for $n$-gram languages, and the same techniques can be used in the probabilistic setting with minor changes.
>
> ""Assuming #1 is fixed…A more accurate and direct title would be "State-space models can express n-gram models".""
> - The title has been updated to “State-space models can express n-gram languages”, which reflects the theorem that we establish. It is slightly unclear what it means for one model to express another model, so we have opted for this variant. It is more common to say that a model can express a language (rather than express another model).
>
> ""There is no discussion about the sharpness of the results. While it is interesting that the author's main theorem allows of any embedding dimension... If the authors could provide some analysis on (sub)optimality, the paper would be more complete in its analysis.
> - We have added a paragraph in Section 6 about tighter bounds on the memorization capacity of state space models following [1]. We expect the proofs of Theorem 3.1 in [1] can be adapted to construct a state space model with two hidden layers and O(K) parameters that can memorize K input-output pairs. We also adapt the proof of Theorem 3.3 in [1], which contains the sharpness results showing that a certain capacity is needed for their representation, to the setting of state space models in Proposition A.8. We show that when the dimension of the output vectors are 1, Proposition 4.5 is optimal for state space models with a single hidden layer. Our proof uses the number of linear pieces in the function to show that those state space models needs at least K neurons to memorize K input-output pairs. Regarding the embedding dimension $e$, we have added a comment in Section 5.1 that the theoretical construction is valid for any $e$, but that stochastic gradient descent does not converge when $e$ is very small.
>
> ""Summarizing, the authors would benefit from referring to [2]… The current paper seems to give a single construction without much context as to how close to optimal it may be.""
> - All of our key definitions and results have been rephrased to align with the probabilistic setting from [2] (see Definition 3.4, Definition 3.5, Theorem 4.1 and Definition 4.2). In Section 6, we have added a discussion about how [1] can be used to obtain tighter bounds on the memorization capacity of state space models with two hidden layers. We added Proposition A.8, which shows Proposition 4.5 is optimal in certain settings. In Theorem 5.1 of [2], it is stated that the hidden dimension has size $|V|^n$, but from their discussion it is not clear to us how this can be mitigated.
>
> ""Notational Feedback: The use of overlines and underlines ... is unclear and should be revised for clarity.""
> - We use the notation $\mathcal{L}$ to denote the language; the overline has been removed throughout the paper. Likewise, the notation for embedding has been updated to E; the overline has been removed. The notation $\overline{f}$ has also been removed. To avoid confusion, the softmax function has been written explicitly as “softmax”, instead of using the notation $\overline{\sigma}$. We note that the symbol $\underline{w}$ always refers to a sequence of words, although there are vectors such as $f(\underline{w})$ which are obtained from it.
>
> ""Expository Feedback: The example following Definition 3.1 could be explained more directly.""
> - The example of the n-gram model in Section 3 has been shortened, and a diagram has been added to clearly illustrate how sentences in this language are formed. We have also added a paragraph in the Appendix describing how the sentences are obtained through word substitutions. The table consisting of a subset of all n-gram rules has been removed from Section 3. We keep the full table in Appendix B, to show that there are 145 $n$-gram rules in $\mathcal{P}$; this fact is used in the experimental section.
>
> [1] Chulhee Yun, Suvrit Sra, and Ali Jadbabaie. Small relu networks are powerful memorizers: a tight analysis of memorization capacity.
> [2] Anej Svete Ryan Cotterell. Transformers Can Represent n-gram Language Models

---

> ### Author Response · Authors · 2025-01-14
>
> Dear Reviewer 4zFd,
>
> We sincerely appreciate your valuable feedback, which has helped us improve the quality of our manuscript. As there are only a few days left in the discussion period, we wanted to check if you have any remaining concerns. If so, we would be glad to have further discussions.
>
> Thank you once again for your time and insights. We look forward to hearing from you.

---

> > ### Comment · Reviewer_4zFd · 2025-01-14
> > **Some more revision please**
> >
> > Hi, thank you for incorporating the probabilistic model. However, there are still some issues:
> >
> > 1) Definition 3.5 is either too weak or else terminology should be revised. You are only comparing the language model with the n-gram model on the support of the n-gram model. You have no control of what f is doing on strings not in your language, and there can be many. Such a measure should not be called epsilon-equivalence.
> >
> > Else, you could follow [2] and consider approximation on arbitrary inputs and the current proof will go through (with the set of all valid strings being all possible strings).
> >
> > 2) I requested that the claim that the embedding dimension can be arbitrary (including dimension=1) be clarified a bit further, since it is somewhat surprising (all other constructions scale with vocab size or n). Please do so since I did not see it addressed.
> >
> > Also it's not that clear what Fig 1 and Fig 3 are really saying. I imagine you are doing something similar to [2] in which by having a large enough hidden dimension, you can just form a one-hot representation of all possible n-grams (or something similar). The nilpotency of A seems to involve the fact that you are shifting around the representations from previous words (moving them to higher index slots). Saying something along those lines would be a helpful supplement to the cartoon images, which are too vague to offer any insight.

---

> ### Author Response · Authors · 2025-01-16
>
> We thank the reviewer for their detailed feedback, and their quick response. We have updated the definitions and incorporated the suggested changes; please see below for more details. Kindly let us know if you have any more concerns for us to address. We look forward to hearing from you and thank you again for your time and insights.
>
> ------
> “Definition 3.5 is either too weak or else terminology should be revised. You are only comparing the language model with the n-gram model on the support of the n-gram model. You have no control of what f is doing on strings not in your language, and there can be many. Such a measure should not be called epsilon-equivalence. Else, you could follow [2] and consider approximation on arbitrary inputs and the current proof will go through (with the set of all valid strings being all possible strings).”
> - The terminology has been revised; instead of saying that two languages are “epsilon-equivalent”, we say that the two languages are “\epsilon-equivalent when restricted to L’’. This terminology clearly reflects the fact that the two language models can differ on input sequences that are not present in the language $L$. We have updated the definition of language model in Section 3.1, so it can take an arbitrary sequence of inputs. We have also updated the definition of an $n$-gram language in Section 3.2, so that it outputs an unknown token if it receives an input whose last $n-1$ words do not occur in its list of $n$-gram rules. While [2] considers all possible strings, the conventions we adopt are closer to those used in empirical work, where the number of $n$-gram rules specified is typically orders of magnitude smaller than the total number of possible strings.
>
> “I requested that the claim that the embedding dimension can be arbitrary (including dimension=1) be clarified a bit further, since it is somewhat surprising (all other constructions scale with vocab size or n). Please do so since I did not see it addressed.”
> - We have added a few sentences in the last paragraph of Section 4.3 discussing the embedding dimension, as it corresponds to the dimension $p$ of the input vector in Proposition 4.5. The key observation is that Lemma A.3 holds generically, even when $p=1$, and the bias term is then chosen to ensure the linear independence of the hidden state vectors. We note that our SSM model may be less expressive if the bias term is not used, and restrictions on $p$ and $e$ could be required in that setting. We have also added a comment in the last paragraph of Section 4.1, referring the reader to the discussion following Proposition 4.5 about why $e$ can be arbitrary.
>
> “Also it's not that clear what Fig 1 and Fig 3 are really saying. I imagine you are doing something similar to [2] in which by having a large enough hidden dimension, you can just form a one-hot representation of all possible n-grams (or something similar). The nilpotency of A seems to involve the fact that you are shifting around the representations from previous words (moving them to higher index slots). Saying something along those lines would be a helpful supplement to the cartoon images, which are too vague to offer any insight.”
> - For Figure 1, we note that while hidden state embeddings encode $n$-gram rules, they are not one-hot vectors. We have updated the figure with a cluster plot of these vectors, after they have projected to two-dimensional space. This cluster plot clearly illustrates how the hidden state embeddings correspond to the $n$-grams (see also Figure 4 for a more detailed cluster plot). For Figure 3, we have added a note stating that the arrows, which show the action of the nilpotent matrix, map each neuron to a neuron with a larger index. Its key purpose is to show diagrammatically why all inputs prior to $x_{T-2}$ do not influence the output $y_T$, and this is evident from the arrows.

---

> > ### Comment · Reviewer_4zFd · 2025-01-17
> > **Notational clarification**
> >
> > Thank you for the changes. I was just skimming over things one last time and notice I still don't understand some notation. Proposition 4.5 and Lemma A.2:
> >
> > "each vector x_i = (x_{i,1}, · · · , x_{i,n}) consists of n components with x_{i,j} ∈ R^p"
> >
> > Not sure I understand. Isn't x_i a vector of lengh n, how is x_{i,j} in R^p?
> >
> > Optional suggestion: The authors may wish to use one alphabet for time indices and another for component indices (e.g. abc vs ijk vs Greek).

---

> ### Author Response · Authors · 2025-01-18
> **Updates to Proposition 4.5 and Lemma A.2.**
>
> We thank the reviewer for drawing our attention to this point. We have updated Proposition 4.5 and Lemma A.2 accordingly; please see below for more details. Please do let us know if you have any other concerns for us to address.
>
> "" Proposition 4.5 and Lemma A.2: "each vector $x_i = (x_{i,1}, · · · , x_{i,n})$ consists of n components with $x_{i,j} ∈ \mathbb{R}^p$
> Not sure I understand. Isn't $x_i$ a vector of length n, how is $x_{i,j} \in \mathbb{R}^p$? ""
>
> We have updated Proposition 4.5 and Lemma A.2, to state clearly that each $x_i$ is an **input sequence** (as opposed to a vector). We also state that **$x_i$ consists of $n$ vectors** (instead of "consists of $n$ components). This should clarify any confusion in the notation.

---

### Author Response · Authors · 2025-01-19
**Summary of changes**

Dear Reviewers,

We deeply appreciate the time and effort you have spent reviewing our manuscript, and your feedback has helped us improve the exposition. We made many revisions to our manuscripts to address them, and the main changes are summarized below.
- A probabilistic framework: We have updated the definitions of n-gram languages to adopt a probabilistic model. We introduced the concept of $\epsilon$-equivalence between language models when "restricted to L", which is used in our formulation of the key results.
- Added figures and explanations of the key proofs: We have added Figures 1, 2 and 3 to help explain the proof of the key results with examples, along with an explanation of the proofs of the key results Proposition 4.5 and Proposition 4.6. We have also updated the notation to remove ambiguities, and included detailed examples in Sections 3 and 4 to help clarify the notation.
- Discussion: We have added material in Section 6 and the Appendix discussing the tightness of memorization bounds. We have also added a paragraph in Section 5 elaborating on comparisons with attention-based models.
- Experiments: We updated the experiments to specify that there are 145 valid n-grams in the language, and aligned with our theoretical results by setting the number of hidden neurons in the state space model to also be 145.

We again express our gratitude for your valuable feedback. Please let us know if there is anything else that we can clarify.

---

### Decision · Action_Editor_W1YX · 2025-02-02

**Recommendation:** Accept as is

**Comment:**

The paper provides a proof by construction on how an SSM can encode a n-th order Markov chain. The construction seems valid, and the result adds to our theoretical understanding of what these architectures can do. The work could have been improved with strong empirical exploration, but I think that is not for acceptance.

**Audience:**

The paper will be of interest to the growing community interested in understanding SSM architectures and their limitations. There is a growing body of works looking at the expressivity properties of these models, and this work fits well and is complementary to those works.
Potentially this work will impact less the more applied part of the community as the result does not directly translates into explicit recommendations in practice.

**Claims And Evidence:**

The paper focuses on SSM architectures, and shows that they can represent an order-n Markov chain. The result is theoretical, the paper provides a proof by construction. While I have not checked the proof carefully myself, based on the reviewer's feedback I believe to be correct. The paper also contains limited empirical evidence, showing that gradient descent can actually discover such solution.

---

> ### Author Response · Authors · 2025-02-20
> **Camera-ready version**
>
> We thank you for your feedback, and thank all three reviewers for their detailed suggestions. We have uploaded the camera-ready pdf, and added a link to our code repository.